# Selective Classification Under Distribution Shifts

**Hengyue Liang**                                                   *liang656@umn.edu*
*Department of Electrical and Computer Engineering*
*University of Minnesota*

**Le Peng**                                                         *peng0347@umn.edu*
*Department of Computer Science and Engineering*
*University of Minnesota*

**Ju Sun**                                                          *jusun@umn.edu*
*Department of Computer Science and Engineering*
*University of Minnesota*

**Reviewed on OpenReview:** *https: // openreview. net/ forum? id= dmxMGW6J7N*

## Abstract

In selective classification (SC), a classifier abstains from making predictions that are likely to be wrong to avoid excessive errors. To deploy imperfect classifiers—either due to intrinsic statistical noise of data or for robustness issue of the classifier or beyond—in high-stakes scenarios, SC appears to be an attractive and necessary path to follow. Despite decades of research in SC, most previous SC methods still focus on the ideal statistical setting only, i.e., the data distribution at deployment is the same as that of training, although practical data can come from the wild. To bridge this gap, in this paper, we propose an SC framework that takes into account distribution shifts, termed *generalized selective classification*, that covers label-shifted (or out-of-distribution) and covariate-shifted samples, in addition to typical in-distribution samples, *the first of its kind* in the SC literature. We focus on non-training-based confidence-score functions for generalized SC on deep learning (DL) classifiers, and propose two novel margin-based score functions. Through extensive analysis and experiments, we show that our proposed score functions are more effective and reliable than the existing ones for generalized SC on a variety of classification tasks and DL classifiers. The code is available at `https://github.com/sun-umn/sc_with_distshift`.

## 1 Introduction

In practice, classifiers almost never have perfect accuracy. Although modern classifiers powered by deep neural networks (DNNs) typically achieve higher accuracy than the classical ones, they are known to be unrobust: perturbations of inputs that are inconsequential to human decision making can easily alter DNN classifiers' predictions (Carlini et al., 2019; Croce et al., 2020; Hendrycks & Dietterich, 2018; Liang et al., 2023), and more generally, shifts in data distribution in deployment from that in training often cause systematic classification errors. These classification errors, regardless of their source, are rarely acceptable for high-stakes applications, such as disease diagnosis in healthcare.

To achieve minimal and controllable levels of classification error so that imperfect and unrobust classifiers can be deployed for high-stakes applications, a promising approach is *selective classification* (SC): samples that are likely to be misclassified are selected, excluded from prediction, and deferred to human decision makers, so that the classification performance on the remaining samples reaches the desired level (Chow, 1970; Franc et al., 2023a; Geifman & El-Yaniv, 2017). For example, by flagging and passing uncertain patient cases that it tends to mistake on to human doctors, an intelligent medical agent can make confident and correct diagnoses for the rest. This "conservative" classification framework not only saves doctors' efforts, but also avoids liability due to the agent's mistakes.

Consider a multiclass classification problem with input space $\mathcal{X} \in \mathbb{R}^n$, label space $\mathcal{Y} = \{1, \ldots, K\}$, and training distribution $\mathcal{D}_{\mathcal{X}, \mathcal{Y}}$ on $\mathcal{X} \times \mathcal{Y}$. For any classifier $f : \mathcal{X} \to \mathcal{Y}$, there are many potential causes of classification errors. In this paper, we focus on three types of errors that are commonly encountered in practice and are studied extensively, but mostly separately, in the literature.

- `Type A errors`: errors made on *in-distribution* (In-D) samples, i.e., those samples drawn from $\mathcal{D}_{\mathcal{X}, \mathcal{Y}}$. These are classification errors discussed in typical statistical learning frameworks (Mohri et al., 2018);
- `Type B errors`: errors made on *label-shifted* samples, i.e., those samples with groundtruth labels not from $\mathcal{Y}$. Since $f$ assigns labels from $\mathcal{Y}$ only, it always errs on these samples;
- `Type C errors`: errors made on *covariate-shifted* samples, i.e., samples drawn from a different input distribution $\mathcal{D}'_{\mathcal{X}}$ where $\mathcal{D}'_{\mathcal{X}} \neq \mathcal{D}_{\mathcal{X}}$ but with groundtruth labels from $\mathcal{Y}$.

It is clear that in practical deployment of classifiers, samples can come from the wild, and hence `Type A`, `Type B` and `Type C` errors can coexist. In order to ensure the reliable deployment of classifiers in high-stakes applications, we must control the three types of errors, *jointly*. Unfortunately, previous research falls short of a unified treatment of these errors. Classical SC (Chow, 1970) focuses on rejecting samples that cause In-D errors (`Type A`), whereas the current *out-of-distribution (OOD) detection* research (Yang et al., 2021; Park et al., 2023) focuses on detecting label-shifted samples (`Type B`). Although Hendrycks & Gimpel (2016); Granese et al. (2021); Xia & Bouganis (2022); Kim et al. (2023) have advocated the simultaneous detection of samples that cause `Type A` and `Type B` errors, their approaches still treat the problem as consisting of two *separate* tasks, reflected in their *separate and independent* performance evaluation on OOD detection and SC. Regarding the challenge posed by `Type C` errors, existing work (Hendrycks & Dietterich, 2018; Croce et al., 2020) focuses primarily on obtaining classifiers that are more robust to covariate shifts, not on rejecting potentially misclassified samples due to covariate shifts—the latter, to the best of our knowledge, has not yet been explicitly considered, not to mention joint rejection together with `Type A` and `Type B` errors.

In this paper, our goal is to close the gap and consider, *for the first time*, rejecting all three types of errors in a unified framework. For brevity, we use the umbrella term *distribution shifts* to cover both label shifts and covariate shifts, which are perhaps the most commonly seen types of distribution shifts, with the caveat that practical distribution shifts can also be induced by other sources. So, we call the unified framework considered in this paper *selective classification under distribution shifts*, or *generalized selective classification*. Another key desideratum is practicality. With the increasing popularity of foundation models and associated downstream few-shot learners (Brown et al., 2020; Radford et al., 2021; Yuan et al., 2021), accessing massive original training data becomes increasingly more difficult. Moreover, there are numerous high-stakes domains where training data are typically protected due to privacy concerns, such as healthcare and finance. These applied scenarios call for *SC strategies that can work with given pretrained classifiers and do not require access to the training data*, which will be our focus in this paper. **Our contributions** include:

- We advocate a new SC framework, *generalized selective classification*, which rejects samples that could cause `Type A`, `Type B` and `Type C` errors *jointly*, to improve classification performance over the non-rejected samples. With careful review and reasoning, we argue that generalized SC covers and unifies the scope of the existing OOD detection and SC, if *the goal is to achieve reliable classification on the selected samples*. (Sections 2.3 and 2.4)
- Focused on non-training-based (or post-hoc) SC settings, we identify a critical scale-sensitivity issue of several SC confidence scores based on softmax responses (Section 3.1) which are popularly used and reported to be the state-of-the-art (SOTA) methods in the existing SC literature (Geifman & El-Yaniv, 2017; Feng et al., 2023).
- We propose two confidence scores based on the raw logits (v.s. the normalized logits, i.e., softmax responses), inspired by the notion of margins (Section 3.2). Through careful analysis (Section 3.3) and extensive experiments (Section 4), we show that our margin-based confidence scores are more reliable for generalized SC on various dataset-classifier combinations, even under moderate distribution shifts.

## 2 Technical background and related work

### 2.1 Selective classification (SC)

Consider a multiclass classification problem with input space $\mathcal{X} \in \mathbb{R}^n$, label space $\mathcal{Y} = \{1, \ldots, K\}$, and data distribution $\mathcal{D}_{\mathcal{X}, \mathcal{Y}}$ on $\mathcal{X} \times \mathcal{Y}$. A selective classifier $(f, g)$ consists of a predictor $f : \mathcal{X} \to \mathbb{R}^K$ and a selector $g : \mathcal{X} \to \{0, 1\}$ and works as follows:

$$(f, g)(\boldsymbol{x}) \triangleq \begin{cases} f(\boldsymbol{x}) & \text{if } g(\boldsymbol{x}) = 1, \\ \text{abstain} & \text{if } g(\boldsymbol{x}) = 0 \end{cases}, \tag{1}$$

for any input $\boldsymbol{x} \in \mathcal{X}$. Typical selectors $g$ take the form:

$$g_{s,\gamma}(\boldsymbol{x}) = \mathbb{1}[s(\boldsymbol{x}) > \gamma], \tag{2}$$

where $s(\boldsymbol{x})$ is a *confidence-score* function, and $\gamma$ is a tunable threshold for selection.

### 2.2 Prior work in SC

For a given selective classifier $(f, g_{s,\gamma})$, its SC performance is often characterized by two quantities:

$$\begin{aligned} (\textbf{coverage}) \ \phi_{s,\gamma} &= \mathbb{E}_{\mathcal{D}_{\mathcal{X},\mathcal{Y}}}[g_{s,\gamma}(\boldsymbol{x})], & \text{(higher the better)}, \\ (\textbf{selection risk}) \ R_{s,\gamma} &= \mathbb{E}_{\mathcal{D}_{\mathcal{X},\mathcal{Y}}}[\ell(f(\boldsymbol{x}), y)g_{s,\gamma}(\boldsymbol{x})]/\phi_{s,\gamma}, & \text{(lower the better)}, \end{aligned} \tag{3}$$

Because a high coverage typically comes with a high selection risk, there is always a need for risk-coverage tradeoff in SC. Most of the existing work considers $\ell$ to be the standard 0/1 classification loss (Chow, 1970; El-Yaniv et al., 2010; Geifman et al., 2018), and we also follow this convention in this paper. A classical cost-based formulation is to optimize the risk-coverage (RC) tradeoff (Chow, 1970)

$$\min_{f, g_{s,\gamma}} \ \mathbb{E}_{\mathcal{D}_{\mathcal{X},\mathcal{Y}}}[\ell(f(\boldsymbol{x}), y)g_{s,\gamma}(\boldsymbol{x})] + \varepsilon\mathbb{E}_{\mathcal{D}_{\mathcal{X},\mathcal{Y}}}[1 - g_{s,\gamma}(\boldsymbol{x})] \quad \equiv \quad \min_{f, g_{s,\gamma}} \ R_{s,\gamma}\phi_{s,\gamma} - \varepsilon\phi_{s,\gamma}, \tag{4}$$

where $\varepsilon \in [0, 1]$ is the cost of making a rejection. The optimal selective classifier for this formulation is (Chow, 1970; Franc et al., 2023a):

$$f^* = \arg\min_{\widehat{y} \in \mathcal{Y}} \sum_{y \in \mathcal{Y}} p(y|\boldsymbol{x})\ell(\widehat{y}, y), \quad \text{and} \quad g^* = \mathbb{1}[-\min_{\widehat{y} \in \mathcal{Y}} \sum_{y \in \mathcal{Y}} p(y|\boldsymbol{x})\ell(\widehat{y}, y) > -\varepsilon], \tag{5}$$

where $f^*$ is the Bayes optimal classifier and depends on the posterior probabilities $p(y|\boldsymbol{x})$ for all $y \in \mathcal{Y}$, which are hard to obtain in practice. Moreover, solutions to two constrained formulations for the RC tradeoff,

$$\min_{f, g_{s,\gamma}} R_{s,\gamma}, \ \text{s.t.} \ \phi_{s,\gamma} \geq \omega \quad \text{and} \quad \max_{f, g_{s,\gamma}} \phi_{s,\gamma}, \ \text{s.t.} \ R_{s,\gamma} \leq \lambda, \tag{6}$$

also depend on the posterior probabilities (Pietraszek, 2005; Geifman & El-Yaniv, 2017; Franc et al., 2023a; El-Yaniv et al., 2010).

**Training-based scores** Due to the intractability of true posterior probabilities in practice, many previous methods focus on learning effective confidence-score functions from training data. They require access to training data and learn parametric score functions, often under cost-based/constrained formulations and their variants for the RC tradeoff. This learning problem can be formulated together with (Chow, 1970; Pietraszek, 2005; Grandvalet et al., 2008; El-Yaniv et al., 2010; Cortes et al., 2016; Geifman & El-Yaniv, 2019; Liu et al., 2019; Huang et al., 2022; Gal & Ghahramani, 2016; Lakshminarayanan et al., 2017; Geifman et al., 2018; Maddox et al., 2019; Dusenberry et al., 2020; Lei, 2014; Villmann et al., 2016; Corbière et al., 2019) or separately from training the classifier (Jiang et al., 2018; Fisch et al., 2022; Franc et al., 2023a). However, Feng et al. (2023) has recently shown that these training-based scores do not outperform simple non-training-based scores described below.

---

**Algorithm 1** Non-training-based selective classification

---

**Require:** A pretrained classifier $f$; a score function $s$; a small calibration dataset $Z^{cali} \sim_{iid} \mathcal{D}^{cali}_{\mathcal{X},\mathcal{Y}}$
  1: $\forall (\boldsymbol{x}_i, y_i) \in Z^{cali}$, compute $s(\boldsymbol{x}_i)$ and $\ell(f(\boldsymbol{x}_i), y_i)$
  2: Determine a threshold $\gamma$ according to the coverage or selection-risk target
  3: Deploy the selector $g_{s,\gamma}$ based on Eq. (2).

---

**Manually designed (non-training-based) scores**  This family works with any given classifier and does not assume access to the training set. This is particularly attractive when it comes to modern pretrained large DNN models, e.g., CLIP (Radford et al., 2021), Florence (Yuan et al., 2021), and GPTs (Brown et al., 2020), for which obtaining the original training data and performing retraining are prohibitively expensive, if not impossible, to typical users. Algorithm 1 shows a typical use case of SC with non-training-based scores. Different confidence scores have been proposed in the literature. For example, for support vector machines (SVMs), confidence margin (the difference of the top two raw logits) has been used as a confidence score (Fumera & Roli, 2002; Franc et al., 2023a); see also Section 3.2. For DNN models, *which is our focus*, confidence scores are popularly defined over the *softmax responses* (SRs). Assume that $\boldsymbol{z} \in \mathbb{R}^K$ contains the raw logits (RLs) and $\sigma$ is the softmax activation. The following three confidence-score functions

$$SR_{\max}(\boldsymbol{z}) \triangleq \max_i \sigma(z_i), \quad SR_{\text{doctor}}(\boldsymbol{z}) \triangleq 1 - 1/\|\sigma(\boldsymbol{z})\|_2^2, \quad SR_{\text{ent}}(\boldsymbol{z}) \triangleq \sum_i \sigma(z_i) \log \sigma(z_i), \quad (7)$$

are popularly used in recent work, e.g., Feng et al. (2023); Granese et al. (2021); Xia & Bouganis (2022). Although simple, $SR_{\max}$ can easily beat existing training-based methods (Feng et al., 2023). On the other hand, these SR-based score functions generally follow the plug-in principle by assuming that SRs approximate posterior probabilities well (Franc et al., 2023a). Unfortunately, this assumption often does not hold in practice, and bridging this approximation gap is a major challenge for confidence calibration (Guo et al., 2017; Nixon et al., 2019). However, Zhu et al. (2022) reveals that recent calibration methods may even degrade SC performance.

## 2.3  SC under distribution shifts: generalized SC

In this paper, we consider SC under distribution shifts, or *generalized selective classification*. Shifts between training and deployment distributions are common in practice and can often cause performance drops in deployment (Quinonero-Candela et al., 2008; Rabanser et al., 2019; Koh et al., 2021), raising reliability concerns for high-stakes applications in the real world. In this paper, we use the term *distribution shifts* to cover both covariate and label shifts—perhaps the most prevalent forms of distribution shifts (see the beginning of Section 1 for their definitions)—jointly. Although the basic set-up for our generalized SC framework remains the same as that of Eqs. (1) and (2), we need to modify the definitions for selection risk and coverage in Eq. (3) to take into account potential distribution shifts:

$$(\textbf{coverage}) \ \phi_{s,\gamma} = \mathbb{E}_{\mathcal{D}'_{\mathcal{X},\mathcal{Y}'}}[g_{s,\gamma}(\boldsymbol{x})], \quad \text{and} \quad (\textbf{selection risk}) \ R_{s,\gamma} = \mathbb{E}_{\mathcal{D}'_{\mathcal{X},\mathcal{Y}'}}[\ell(f(\boldsymbol{x}), y) g_{s,\gamma}(\boldsymbol{x})]/\phi_{s,\gamma}, \quad (8)$$

where $\mathcal{D}_{\mathcal{X},\mathcal{Y}}$ is the original data distribution, $\mathcal{D}'_{\mathcal{X},\mathcal{Y}'}$ is the shifted distribution—$\mathcal{Y}'$ may not be the same as $\mathcal{Y}$ due to potential label shifts.[1]

**Out-of-distribution (OOD) detection as a weak form of generalized SC**  The goal of OOD detection is to detect and exclude OOD samples (Yang et al., 2021). An ideal OOD detector $G(\boldsymbol{x})$ should perfectly separate In-D and OOD samples:

$$G(\boldsymbol{x}) = \begin{cases} 0 \ (\text{i.e., excluded}) & \text{if } \boldsymbol{x} \sim \mathcal{D}^{\text{OOD}}_{\mathcal{X}} \\ 1 \ (\text{i.e., kept}) & \text{if } \boldsymbol{x} \sim \mathcal{D}_{\mathcal{X}} \end{cases}, \quad \text{which is often realized as} \quad G(\boldsymbol{x}) = \mathbb{1}[s_{\text{OOD}}(\boldsymbol{x}) > \gamma]. \quad (9)$$

---

[1]We assume no *outliers* in generalized SC—samples that do not follow any specific statistical patterns—during deployment, i.e., they are already detected and removed after separate data preprocessing steps. This allows us to properly define the coverage and selection risk.

---

**Algorithm 2** Typical OOD detection pipeline (e.g., Sun et al. (2021))

---

**Require:** An OOD score function $s_{OOD}$; an In-D calibration dataset $X^{in} \sim_{iid} \mathcal{D}_{\mathcal{X}}$ and an OOD calibration dataset $X^{ood} \sim_{iid} \mathcal{D}_{\mathcal{X}}^{OOD}$

1: $\forall \boldsymbol{x}_i \in X^{in}$ and $\forall \boldsymbol{x}_j \in X^{ood}$, compute $s_{OOD}(\boldsymbol{x}_i)$ and $s_{OOD}(\boldsymbol{x}_j)$.
2: Compute a threshold $\gamma_{\text{OOD}}$ using Eq. (9) by problem-specific target requirements, e.g., a target TPR (true positive rate) value.
3: Deploy the OOD detector according to Eq. (9).

---

Here, $s_{\text{OOD}}(\cdot)$ is a confidence-score function indicating the likelihood that the input is an In-D sample, and $\gamma$ is again a tunable cutoff threshold. Although by the literal meaning of OOD both covariate and label shifts are covered by $\mathcal{D}_{\mathcal{X}}^{\text{OOD}}$, the literature on OOD detection focuses mainly on detecting *label-shifted* samples, i.e., covariate-shifted $\mathcal{D}_{\mathcal{X}}^{\text{OOD}}$ induced by label shifts (Liu et al., 2020; Sun et al., 2021; Wang et al., 2022; Sun et al., 2022). OOD detection is commonly motivated as an approach to achieving reliable predictions: under the assumption that $\mathcal{D}_{\mathcal{X}}^{\text{OOD}}$ is induced by label shifts only, any OOD samples will cause misclassification and hence should be excluded—clearly aligned with the goal of SC. Algorithm 2 shows the typical use case of OOD (label-shift) detectors, and its similarity to SC shown in Algorithm 1 is self-evident. However, OOD detection clearly aims for less than generalized SC in that: (1) even if the OOD detection is perfect, misclassified samples—either as In-D or due to distribution shifts—by imperfect classifiers are not rejected, and (2) practical OOD detectors may fail to perfectly separate In-D and OOD samples, OOD detected but correctly classified In-D samples are still rejected, hurting the classification performance on the selected samples; see Appendix C for an illustrative example. Therefore, if we are to achieve reliable predictions by excluding samples that are likely to cause errors, we should directly follow the generalized SC instead of the OOD detection formulation.

**Other related concepts** Besides OOD detection, OOD generalization focuses on correctly classifying In-D and covariate-shifted samples, without considering prediction confidence and selection to improve prediction reliability; open-set recognition (OSR) focuses on correctly classifying In-D samples, as well as flagging label-shifted samples; see Geng et al. (2020) for a comprehensive review. In contrast, generalized SC covers all In-D, label-shifted, and covariate-shifted samples, the widest coverage compared to these related concepts, and targets the most practical and pragmatic metric—classification performance on the selected samples.

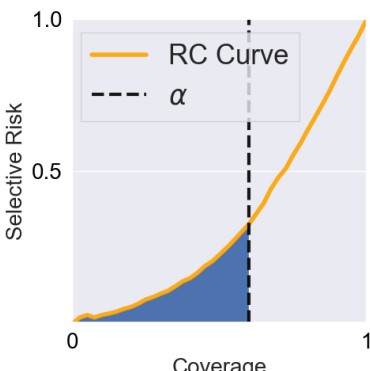

Figure 1: Visualization of the *normalized AURC-α*—the area in blue divided by the coverage value $\alpha$.

**Prior work on SC with distribution shifts** Although the existing literature on SC is rich (Zhang et al., 2023), research work that considers SC with potential distribution shifts is very recent and focuses only on *label shifts*: Xia & Bouganis (2022); Kim et al. (2023) perform In-D SC and OOD (label shift) detection together with a confidence score that combines an SC score and an OOD score, but they still evaluate the performance of In-D SC and OOD detection *separately*. Müller et al. (2023); Cattelan & Silva empirically show that existing OOD scores are not good enough for SC tasks with covariate/label-shifted samples; Cattelan & Silva proposes ways to refine these scores with the help of additional datasets to optimize performance. Franc et al. (2024) provides theoretical insights on SC with In-D and label-shifted samples. In contrast, we focus on identifying better confidence scores for generalized SC—that covers both In-D and covariate/label-shifted samples and maximizes the utility of the classifier, and unify the evaluation protocol (see Section 2.4).

## 2.4 Evaluation of generalized SC

Since the goal of generalized SC is to identify and exclude misclassified samples, for performance evaluation at a fixed cutoff threshold $\gamma$, it is natural to report the coverage—the portion of samples accepted, and the corresponding selection risk—"accuracy" (taken broadly) on accepted samples. It is clear from Eqs. (1)

and (2) that for a given pair of classifier $f$ and confidence-score function $s$, the threshold $\gamma$ can be adjusted to achieve different risk-coverage (RC) tradeoffs. By continuously varying $\gamma$, we can plot a *risk-coverage (RC) curve* El-Yaniv et al. (2010); Franc et al. (2023a) to profile the SC performance of $(f, s)$ throughout the entire coverage range $\phi_\gamma \in [0, 1]$; see Fig. 1 for an example. Generally, the lower the RC curve, the better the SC performance. To obtain a summarizing metric, it is natural to use the *area under the RC curve* (AURC) (El-Yaniv et al., 2010; Franc et al., 2023a). We note that the RC curve and the AURC are also the most widely used evaluation metrics for classical SC—which is not surprising, as the goal of classical SC aligns with that of generalized SC, although generalized SC also allows distribution shifts.

For typical high-stakes applications, such as medical diagnosis, low selection risks are often prioritized over high coverage levels. So, in addition to RC curves and AURC, we also report several partial AURCs to account for potential different needs—*normalized AURC-$\alpha$*, where $\alpha$ specifies the coverage level, and we normalize the partial area-under-the-curve by the corresponding $\alpha$ so that different partial levels can be cross-compared; see Fig. 1 for illustration.

Note that RC curves, and hence the associated AURCs and normalized AURC-$\alpha$ also, depend on the $(f, s)$ pair. So, if the purpose is to *compare different confidence-score functions*, $f$ should be fixed. Feng et al. (2023) has recently pointed out the abuse of this crucial point in recent training-based SC methods. Thus, it is worth stressing that we *always take and fix pretrained $f$'s* when making the comparison between different score functions.

### 2.5 Few words on implementing Algorithm 1 in practice

In the practical implementation of generalized SC for high-stakes applications after Algorithm 1, it is necessary to select a cutoff threshold $\gamma$ based on a calibration set to meet the target coverage, or more likely the target risk level. However, in this paper, we follow most existing work on SC and do not touch on issues such as how the calibration set should be constructed and how the threshold should be selected—we leave these for future work. Our evaluation here, again, as most existing SC work, is only about the *potential* of specific confidence-score functions for generalized SC, measured by the RC curve, AUPC, and normalized AURC-$\alpha$'s, directly on test sets that consist of In-D, OOD, and covariate-shifted samples.

## 3 Our method—margins as confidence scores for generalized SC

Our goal is to design effective confidence-score functions for generalized SC. Again, our focus is on non-training-based scores that can work on any pretrained classifier $f$ without access to the training data.

### 3.1 Scale sensitivity of SR-based scores

As discussed in Section 2.2, most manually designed confidence scores focus on DNN models and are based on softmax responses (SRs), assuming that SRs closely approximate true posterior probabilities—closing such approximation gaps is the goal of confidence calibration. However, effective confidence calibration remains elusive (Guo et al., 2017; Nixon et al., 2019), and the performance of SR-based score functions is sensitive to the scale of raw logits and hence that of SRs, as explained below.

**A quick numerical experiment** Consider a 4-component mixture-of-Gaussian distribution with means $\boldsymbol{w}_1 = [\sqrt{2}/2, \sqrt{2}/2]^\intercal$, $\boldsymbol{w}_2 = [-\sqrt{2}/2, \sqrt{2}/2]^\intercal$, $\boldsymbol{w}_3 = [-\sqrt{2}/2, -\sqrt{2}/2]^\intercal$, $\boldsymbol{w}_4 = [\sqrt{2}/2, -\sqrt{2}/2]^\intercal$, equal variance $0.15 \times \boldsymbol{I}$, and equal weight $1/4$. If we treat each component of the mixture as a class and consider the resulting 4-class classification problem, it is easy to see that the optimal 4-class linear classifier is $f(\boldsymbol{x}) = [\boldsymbol{w}_1, \boldsymbol{w}_2, \boldsymbol{w}_3, \boldsymbol{w}_4]^\intercal \boldsymbol{x}$, with the decision rule $\arg\max_{j \in \{1,2,3,4\}} \boldsymbol{w}_j^\intercal \boldsymbol{x}$; see Fig. 2 **(a)** for visualization of the data distribution and decision boundaries (i.e., the lines $\boldsymbol{x}_1 = 0$ and $\boldsymbol{x}_2 = 0$). Moreover, this $f(\boldsymbol{x})$ is also a Bayes optimal classifier as well as the maximum a posterior (MAP) classifier, for our particular problem here. Now, given any input $\boldsymbol{x}$, we consider scaled raw logits $\lambda f(\boldsymbol{x})$ for different scale factors $\lambda = 0.1, 1, 2, 4$ and plot the resulting RC curves for $SR_{\max}$, $SR_{\text{doctor}}$, and $SR_{\text{ent}}$, respectively; see Fig. 2 **(b)-(d)**. For reference, we also include the RC curves based on the true posterior probabilities (denoted as $s_{\text{post}}$), which are available for our simple data model here. We can observe that for SR-based functions ($SR_{\max}$, $SR_{\text{doctor}}$, and $SR_{\text{ent}}$),

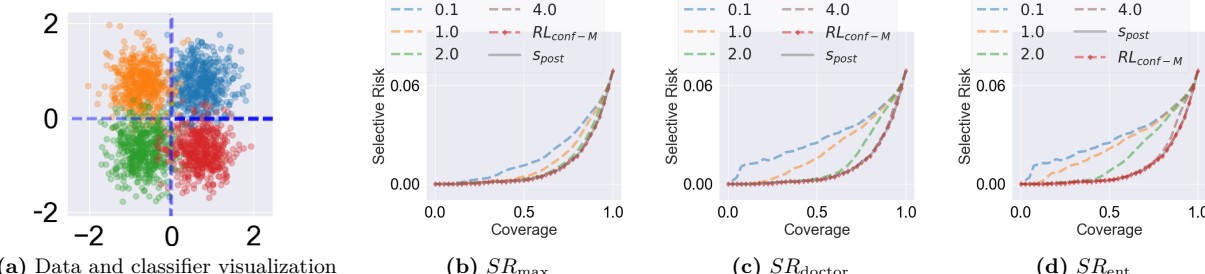

**(a)** Data and classifier visualization  **(b)** $SR_{\max}$  **(c)** $SR_{\text{doctor}}$  **(d)** $SR_{\text{ent}}$

Figure 2: RC curves for **(b)** $SR_{\max}$, **(c)** $SR_{\text{doctor}}$, and **(d)** $SR_{\text{ent}}$, calculated based on scaled (by factor 0.1, 1.0, 2.0, and 4.0, respectively) raw logits from the optimal 4-class linear classifier using data shown in **(a)**. The RC curves for $RL_{\text{conf-M}}$ and $s_{\text{post}}$ are also plotted for reference, where $RL_{\text{conf-M}}$ is one of our proposed confidence-score functions.

their RC curves and hence the associated AURC's vary as $\lambda$ changes, and these curves approach a common curve ($RL_{\text{conf-M}}$, which we will explain below) as $\lambda$ becomes large.

**Why it happens?** The above observations are not incidental. To see why the curves change with respect to $\lambda$, note that for a given test set $\{\boldsymbol{x}_i\}$ and a fixed classifier $f$, the RC curve for any score function $s$ is fully determined by the ordering of $s(\boldsymbol{x}_i)$'s (Franc et al., 2023a). But this ordering is sensitive to the scale of the raw logits for all three SR-based score functions: $SR_{\max}$, $SR_{\text{doctor}}$, and $SR_{\text{ent}}$. Take $s = SR_{\max}$ as an example and consider any sample $\boldsymbol{x}$ with its corresponding raw logits $\boldsymbol{z}$ sorted in descending order (i.e., $z^{(1)} \geq z^{(2)} \geq \cdots$) without loss of generality. Then for any scale factor $\lambda > 0$ applied to $\boldsymbol{z}$, we have the score

$$SR_{\max} \triangleq e^{\lambda z^{(1)}} / \sum_j e^{\lambda z^{(j)}} = 1 / \sum_j e^{\lambda(z^{(j)} - z^{(1)})} = \exp\left(-\log \sum_j e^{\lambda(z^{(j)} - z^{(1)})}\right). \tag{10}$$

This means that the score is determined by all the scaled *logit gaps* $\lambda(z^{(j)} - z^{(1)})$'s. Moreover, due to the inner exponential function, small gaps gain more emphasis as $\lambda$ increases, and all gaps receive increasingly more emphasis as $\lambda$ decreases. Such a shifted emphasis can easily change the order of scores for two data samples, depending on how different their raw logits are distributed. Clearly, $e^{\lambda z^{(1)}} / \sum_j e^{\lambda z^{(j)}} = 1/K$ as $\lambda = 0$. We can also make similar arguments for $SR_{\text{doctor}}$ and $SR_{\text{ent}}$. Next, for the common asymptotic curve as $\lambda \to \infty$, we can show the following (proof is deferred to Appendix B):

**Lemma 3.1.** *Consider the raw logits $\boldsymbol{z}$, and without loss of generality assume that they are ordered in descending order without any ties, i.e., $z^{(1)} > z^{(2)} > \cdots$. We have that as $\lambda \to \infty$,*

$$SR_{\max}(\lambda\boldsymbol{z}) \sim \exp\left(-e^{\lambda(z^{(2)} - z^{(1)})}\right), \quad SR_{\text{doctor}}(\lambda\boldsymbol{z}) \sim 1 - \exp\left(2e^{\lambda(z^{(2)} - z^{(1)})}\right), \quad SR_{\text{ent}}(\lambda\boldsymbol{z}) \sim -e^{\lambda(z^{(2)} - z^{(1)})},$$

*where $\sim$ means asymptotic equivalence. In particular, all the asymptotic functions increase monotonically with respect to $z^{(1)} - z^{(2)}$.*

This implies that the asymptotic RC curve as $\lambda \to \infty$ for all three score functions is fully determined by the score function $z^{(1)} - z^{(2)}$!

**Implications** The sensitivity of the RC curves, and hence of the performance, of these SR-based scores to the scale of raw logits is disturbing. *It implies that one can simply change the overall scale of the raw logits—which does not alter the classification accuracy itself—to claim better or worse performance of an SR-based confidence-score function for selective classification, making the comparison of different SR-based scores shaky.* Unfortunately, between the limiting cases $\lambda \to 0$ and $\lambda \to \infty$, there is no canonical scaling.

## 3.2 Our method: margin-based confidence scores

To avoid the scale sensitivity caused by the softmax nonlinearity, it is natural to consider designing score functions directly over the raw logits. To this end, we revisit ideas in support vector machines (SVMs).

**Margins in SVMs**  In linear SVMs for binary classification, the classifier takes the form $f(\boldsymbol{x}) = \text{sign}(\boldsymbol{w}^\intercal \boldsymbol{x} + b)$ and the confidence in classifying a sample $\boldsymbol{x}$ can be assessed by its distance from the supporting hyperplane (Fumera & Roli, 2002; Franc et al., 2023a): $|\boldsymbol{w}^\intercal \boldsymbol{x} + b|/\|\boldsymbol{w}\|_2$, which is called the *geometric margin*; see Appendix A for a detailed review. We can extend the idea to $K$-class linear SVMs. Following the popular joint multiclass SVM formulation (Crammer & Singer, 2001), we consider a linear classifier $f(\boldsymbol{x}) = \boldsymbol{W}^\intercal \boldsymbol{x} + \boldsymbol{b}$. Here, $\boldsymbol{W}$ and $\boldsymbol{b}$ induce $K$ hyperplanes, and we can define the signed distance of any sample $\boldsymbol{x}$ to the $i$-th hyperplane as: $(\boldsymbol{w}_i^\intercal \boldsymbol{x} + b_i)/\|\boldsymbol{w}_i\|$ ($\boldsymbol{w}_i$ denotes the $i$-th column of $\boldsymbol{W}$ and $b_i$ the $i$-th element of $\boldsymbol{b}$), generalizing the definition for the binary case. However, a single signed distance makes little sense for assessing the classification confidence in multiclass cases, given the typical `argmax` decision rule—e.g., the largest signed distance can be negative. Instead, comparing the distances to all decision hyperplanes seems more reasonable. Thus, we can consider the following *geometric margin* as a confidence-score function:

$$(\boldsymbol{w}_{y'}^\intercal \boldsymbol{x} + b_{y'})/\|\boldsymbol{w}_{y'}\|_2 - \max_{j \in \{1,\ldots,K\} \setminus y'} (\boldsymbol{w}_j^\intercal \boldsymbol{x} + b_j)/\|\boldsymbol{w}_j\|_2 \ , \tag{11}$$

where $y' \in \arg\max_{j \in \{1,\ldots,K\}} (\boldsymbol{w}_j^\intercal \boldsymbol{x} + b_j)/\|\boldsymbol{w}_j\|_2$. In other words, it is the difference between the top two signed distances of $\boldsymbol{x}$ to all $K$ hyperplanes. Intuitively, the larger the geometric margin, the more confident the classifier is in classifying the sample following the largest signed distance—*a clearer winner earns more trust*. Although the interpretation is intuitive, the geometric margin is not popularly used in multiclass SVM formulations, likely due to its non-convexity. Instead, a popular proxy for the geometric margin is the convex *confidence margin*:

$$(\boldsymbol{w}_{y'}^\intercal \boldsymbol{x} + b_{y'}) - \max_{i \in \{1,\ldots,K\} \setminus y'} (\boldsymbol{w}_i^\intercal \boldsymbol{x} + b_i), \tag{12}$$

with the decision rule $y' \in \arg\max_{j \in \{1,\cdots,K\}} \boldsymbol{w}_j^\intercal \boldsymbol{x} + b_j$; see Appendix A. Despite its numerical convenience, the confidence margin loses geometric interpretability compared to the geometric margin, and it can be sensitive to the scaling of $\boldsymbol{w}_j$. We study both margins in this paper.

**Margins in DNNs**  To extend the idea of margins to a DNN classifier $f_{\boldsymbol{\theta}}(\boldsymbol{x})$ parameterized by $\boldsymbol{\theta}$, we view all but the final linear layer as a feature extractor, denoted as $f_{\boldsymbol{\theta}}^e$. So, for each sample $\boldsymbol{x}$, the logit output takes the form $\boldsymbol{z} = \boldsymbol{W}^\intercal f_{\boldsymbol{\theta}}^e(\boldsymbol{x}) + \boldsymbol{b}$, and thus the signed distance of the representation $f_{\boldsymbol{\theta}}^e(\boldsymbol{x})$ to each decision hyperplane in the representation space is: $d_j = (\boldsymbol{w}_j^\intercal f_{\boldsymbol{\theta}}^e(\boldsymbol{x}) + b_j)/\|\boldsymbol{w}_j\|_2 \quad \forall j \in \{1,\ldots,K\}$. Assume sorted signed distances and logits, i.e., $d^{(1)} \geq d^{(2)} \geq \ldots \geq d^{(K)}$ and $z^{(1)} \geq z^{(2)} \geq \ldots \geq z^{(K)}$. The *geometric margin* and the *confidence margin* are defined as

$$RL_{\text{geo-M}} \triangleq d^{(1)} - d^{(2)} \quad \text{and} \quad RL_{\text{conf-M}} \triangleq z^{(1)} - z^{(2)}, \text{respectively.} \tag{13}$$

Note that both $RL_{\text{geo-M}}$ and $RL_{\text{conf-M}}$ are computed using the *raw logits without softmax normalization*; $z$'s and $d$'s may not have the same ordering due to the scale of $\|\boldsymbol{w}_j\|_2$. In fact, $RL_{\text{conf-M}}$ is applied in LeCun et al. (1989) to formulate an empirical rejection rule for a handwritten recognition system, although no detailed analysis or discussion is given on why it is effective. Despite the simplicity of these two notions of margins, we have not found prior work that considers them for SC except for LeCun et al. (1989).

**Scale-invariance property**  An attractive property of margin-based score functions is that their SC performance is *invariant* w.r.t. the scale of raw logits. This is because changing the overall scale of the raw logits does not change the order of scores assigned by either the geometric or the confidence margin. In this regard, margin-based score functions are much more preferred and reliable than SR-based scores for SC. Another interesting point is that the limiting curve depicted in Fig. 2**(b)-(d)** is induced by the confidence margin, as is clear from Lemma 3.1 and the discussion following it.

### 3.3 Analysis of rejection patterns

We continue with the toy example in Section 3.1 to show another major difference between the SR-based and the margin-based score functions—they have *different rejection patterns for given coverage levels*. We will see that margin-based score functions induce favorable rejection patterns and can hence be used for reliable rejection even under moderate covariate shifts. For comparison, we also consider the maximum raw logit (denoted as $RL_{\text{max}}$) to show that a single logit in multiclass classification is not a sensible confidence score.

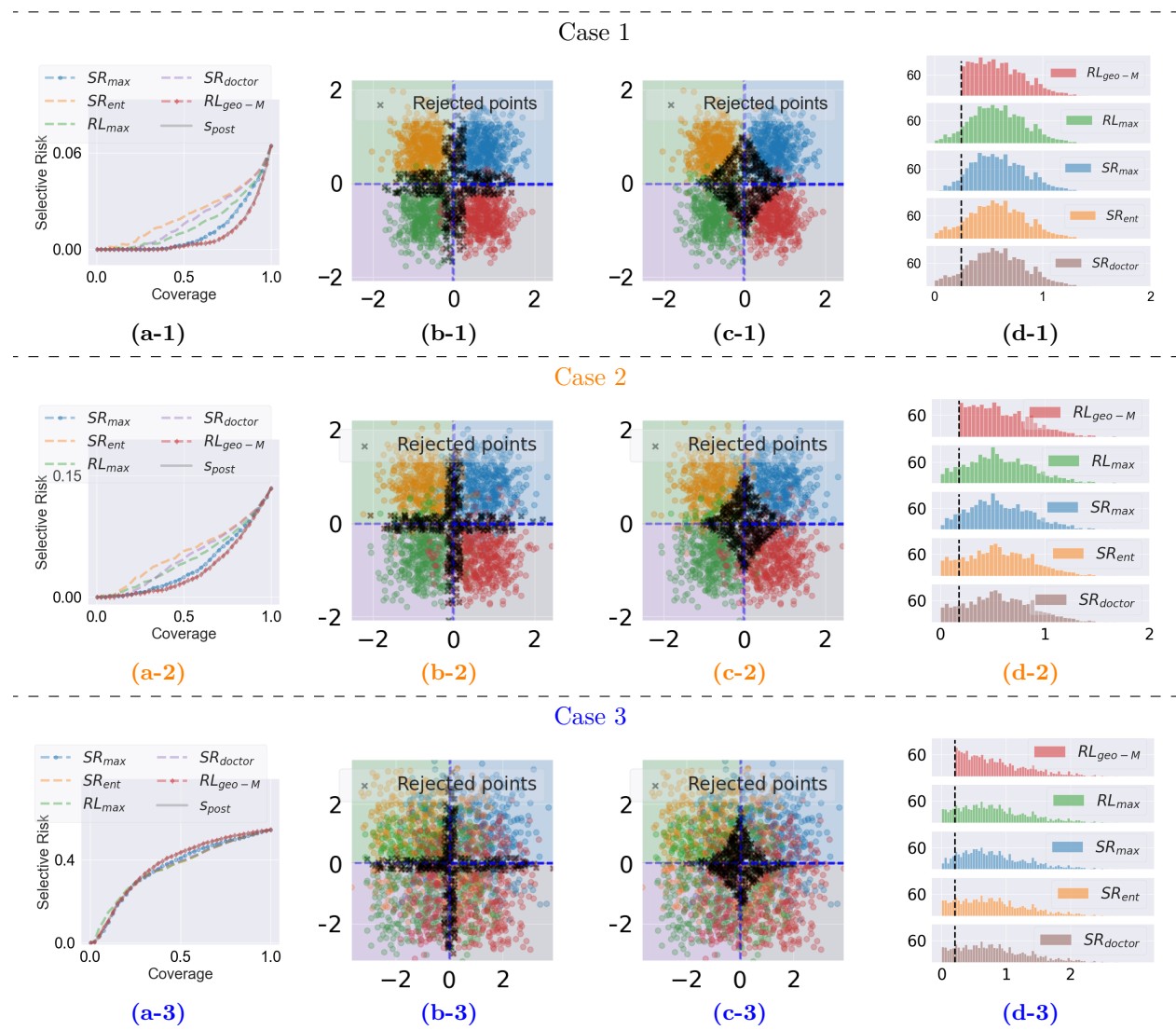

Figure 3: Further analysis of the numerical example in Section 3.1. Case 1, Case 2, and Case 3 correspond to the original dataset in Section 3.1, the dataset after small perturbations, and the dataset after substantial perturbations, respectively. Here, **(a-)**'s are the RC curves achieved by different selection scores; **(b-)**'s are visualizations of the samples (one color per class), decision boundaries (dashed blue line) and the rejected samples (black crosses) at coverage 0.8 by $RL_{\text{geo-M}}$; **(c-)**'s visualize the rejected samples (black crosses) at coverage 0.8 by $SR_{\max}$; and **(d-)**'s present the histogram of the robustness radius of the selected samples in by all score functions.

**Case 1:** We use the same setup as in the numerical experiment in Section 3.1 (see also Fig. 2), and plot in Fig. 3 **(a-1)** the RC curves induced by the various confidence-score functions[2]. It is clear that $RL_{\text{geo-M}}$ performs the best. To better understand the difference between $RL_{\text{geo-M}}$ and other score functions, we study their rejection patterns: we visualize in Fig. 3 **(b-1)&(c-1)** the samples rejected at 0.8 coverage for $RL_{\text{geo-M}}$ and $SR_{\max}$, respectively; see visualization of other score functions in Appendix D, whose rejection patterns are similar to that of $SR_{\max}$. *An iconic feature of $RL_{geo-M}$ is that it prioritizes rejecting samples closer to decision boundaries, whereas SR-based scores prioritize rejecting samples close to the origin.* Conceptually, the former rejection pattern is favorable, as the goal of SC is exactly to reject uncertain samples on which classifier's decisions can be shaky. More precisely, the difference in rejection

---

[2]For the classifier consideblue, $RL_{\text{geo-M}}$ and $RL_{\text{conf-M}}$ have the same SC performance as $\|w\|_2 = 1$.

patterns implies at least two things: (1) $RL_{\text{geo-M}}$ could be advantageous when most classification errors occur near the decision boundaries; (2) $RL_{\text{geo-M}}$ may be superior even when test samples have a moderate level of distribution shifts with respect to training. For example, when the test set has a slightly different $\mathcal{D}_{\mathcal{X}|\mathcal{Y}}$ than the training set (see **Cases 2** & **3** below), mistaken samples due to the shift tend to be close to the decision boundaries and thus can be successfully rejected. Fig. 3 **(d-1)** plots the histograms of the *robustness radii* (i.e., the $\ell_2$ distance of a sample to the closest decision boundary) of selected samples at 0.8 coverage, where the robustness radius quantitatively captures the extent of $\mathcal{D}_{\mathcal{X}|\mathcal{Y}}$ shift SC can tolerate: while the selected samples using $RL_{\text{geo-M}}$ uniformly have nonzero robust radii, all other score functions lead to zero robustness radii for the worst samples, implying sensitivity to $\mathcal{D}_{\mathcal{X}|\mathcal{Y}}$ shifts.[3]

**Case 2:** We keep the same setup as Case 1, except that small perturbations are added on all samples. The perturbations are drawn from a uniform distribution within the interval $[-0.5, 0.5]$ on each dimension of $\mathbb{R}^2$; see Fig. 3 **(b-2)**, where more samples of different classes are intermingled than before the perturbations are added. Although some misclassified samples have moved far into the bulks of other classes, most of them are still close to the decision boundaries. Therefore, $RL_{\text{geo-M}}$ still outperforms other SR-based score functions, as in Fig. 3 **(a-2)**.

**Case 3:** We continue to increase the magnitudes of perturbations and Fig. 3 **(b-3)** illustrates the case where the perturbations are drawn from a uniform distribution within the interval $[-2, 2]$. Now that samples from different classes are well mixed in the 2D space, $RL_{\text{geo-M}}$ is no longer superior when the coverage level is high, as shown in Fig. 3 **(a-3)**. However, we argue that **Case 3** is less concerning in practice—we probably will never consider deploying a classifier that does not work well at all before SC; see the risk achieved at coverage level 1. Instead of relying on an SC strategy, it is more urgent to improve the base classifier in this case.

**Summary:** Using the above examples, we have shown that our proposed margin-based score functions are not sensitive to the scale of the raw logits. When the base classifier is reasonable in classifying in-distribution data samples (i.e., achieving low risks at full coverage), margin-based scores are expected to result in good SC performance, even when test samples have low or moderate distribution shifts, as we show empirically in Section 4 below.

## 4 Experiments

In this section, we experiment with various multiclass classification tasks and recent DNN classifiers to verify the effectiveness of our margin-based score functions for generalized SC.

### 4.1 Comparison with nontraining-based score functions using pretrained models

**Setups** We take different pretrained DNN models in various classification tasks and evaluate SC performance on test datasets composed of In-D and distribution-shifted samples jointly. Specifically, our evaluation tasks include (i) `ImageNet` (Russakovsky et al., 2015), the most widely used testbed for image classification, with a covariate-shifted version `ImageNet-C` (Hendrycks & Dietterich, 2018) composed of synthetic perturbations, and `OpenImage-O` (Wang et al., 2022) composed of natural images similar to `ImageNet` but with disjoint labels, i.e., label-shifted samples; (ii) `iWIldCam` (Beery et al., 2020) test set provides two subsets of animal images taken at different geo-locations, where one is the same as the training set serving as In-D and the other at different locations as a natural covariate-shifted version; (iii) `Amazon` (Ni et al., 2019) test set provides two subsets of review comments by different users, producing In-D and natural covariate-shifted test samples for a language sentiment classification task; (iv) `CIFAR-10` (Krizhevsky et al., 2009), a small image classification dataset commonly used in previous training-based SC works, together with `CIFAR-10-C` (perturbed `CIFAR-10`) and `CIFAR-100` (with disjoint labels from `CIFAR-10`), popularly used covariate-shifted and label-shifted versions of `CIFAR-10`. Tables 1 and 2 summarize the pretrained models and datasets.

**Confidence-score functions for comparison** In addition to $SR_{\max}$, $SR_{\text{doctor}}$ and $SR_{\text{ent}}$ introduced in Eq. (7) and our proposed margin-based scores $RL_{\text{geo-M}}$ and $RL_{\text{conf-M}}$ in Eq. (13), we also consider several

---

[3]The intuition on why our notions of margins work for Type B errors is different: there since $\boldsymbol{x}$ assumes a label outside the known set, we expect no clear winner in the raw logits.

recent post-hoc OOD detection scores[4]: (i) $RL_{max}$: the maximum raw logit (Hendrycks et al., 2019); (ii) Energy: log-sum-exponential aggregation (i.e., smooth approximation to the maximum raw logit) of the raw logits (Liu et al., 2020); (iii) KNN: a score composed of the distances from a test data point to the $k$ nearest neighbors of the training set in the raw logit space (Sun et al., 2022); (iv) ViM—a score composed of the residual of a test sample from the principal components estimated in the feature space prior to the raw logits using training data (Wang et al., 2022); and (v) SIRC—a composite score of the softmax response and OOD detection scores (Xia & Bouganis, 2022). **We note that** KNN, ViM, and SIRC all contain hyperparameters that are determined by the training data. To minimize the gap with our 'nontraining-based' setup, we randomly sample a small number of data points[5] from the In-D test set to tune their hyperparameters, respectively. Also, note that KNN has an additional hyperparameter $k$ that is independent of the statistics of the dataset. Empirically, we find KNN's performance is very sensitive to the choice of $k$, the task, and the classifier. Therefore, in this paper, we use $k = 2$ (the empirical best) by default for KNN and provide an ablation analysis for KNN for each experiment in Appendix H.

Table 1: Summary of the pretrained classifiers used for the various classification tasks

| Task | Model Name | Source | Note |
|---|---|---|---|
| ImageNet | EVA (Fang et al., 2023)
ConvNext (Liu et al., 2022)
VOLO (Yuan et al., 2022)
ResNext (Xie et al., 2017) | `timm`[6] | Top-1 acc. 88.76 %
Top-1 acc. 86.25 %
Top-1 acc. 85.56 %
Top-1 acc. 85.54 % |
| iWildCam | FLYP (Goyal et al., 2023) | Official source code[7] | Ranked $1^{st}$ on `WILDS` (Koh et al., 2021) |
| Amazon | LISA (Yao et al., 2022) | Official source code[8] | Ranked $1^{st}$ on `WILDS` |
| CIFAR & ImageNet | ScNet (Geifman & El-Yaniv, 2019) | PyTorch re-implementation[9] | Training-based SC. |

Table 2: Summary of In-D and distribution-shifted datasets used for our SC evaluation

| Task | In-D (split) | classes - samples | Shift-Cov | samples | Shift-Label | samples |
|---|---|---|---|---|---|---|
| ImageNet | ILSVRC-2012 ('val') | 1000 - 50,000 | ImageNet-C (severity 3)
*All types of corruptions | $50{,}000 \times 19$ | OpenImage-O | 17,256 |
| iWildCam | iWildCam ('id_test') | 178 - 8154 | iWildCam ('ood_test') | 42791 | N/A | N/A |
| Amazon | Amazon ('id_test') | 5 - 46,950 | Amazon ('test') | 100,050 | N/A | N/A |
| CIFAR | CIFAR-10 ('val') | 10 - 10,000 | CIFAR-10-C (severity 3)
*All types of corruptions | $10{,}000 \times 19$ | CIFAR-100 | 10,000 |

**Evaluation metrics** We report both the RC curves and the AURC-$\alpha$ where $\alpha \in \{0.1, 0.5, 1\}$ as discussed in Section 2.4. Note that when plotting the RC curves, we omit $SR_{doctor}$ because it almost overlaps with $SR_{max}$, which is also observed by Xia & Bouganis (2022).

**Results on ImageNet** We show in Fig. 4 the RC curves of the various score functions on the pretrained model **EVA**, for different combinations of subsets of test data, as summarized in Table 3. The most striking is in Fig. 4(c), which collects the results for evaluation on mixup of In-D and label-shifted samples: except for $RL_{geo-M}$, $RL_{conf-M}$ and KNN, the selection risks of other score functions do not follow a monotonic decreasing trend as coverage decreases. As coverage approaches zero, their selection risks spike up, almost to the risk level at full coverage (i.e., error rate on the whole set). This is because the other score functions do not indicate prediction confidence well in this setting and hence fail to sufficiently separate right and wrong

---

[4]In OOD detection, scores are usually dependent on the training data. However, these post-hoc scores can also be applied as nontraining-based SC scores as Algorithm 1, by replacing $\mathcal{D}_{\mathcal{X}}$ and $\mathcal{D}_{\mathcal{X}}^{OOD}$ in Algorithm 2 with $\mathcal{D}_{\mathcal{X},\mathcal{Y}}^{cali}$.

[5]Five times the number of classes in each task from Table 2. We do not sample five points per class, as in practice the calibration set $\mathcal{D}_{\mathcal{X},\mathcal{Y}}^{cali}$ may be imbalanced.

[6]See Table 6 in Appendix E for the model card information to retrieve these `timm` models.

[7]https://github.com/locuslab/FLYP

[8]https://github.com/huaxiuyao/LISA.git

[9]https://github.com/gatheluck/pytorch-SelectiveNet

predictions—during rejection, both right and wrong predictions are rejected indiscriminately. On the other hand, $RL_{\text{geo-M}}$, $RL_{\text{conf-M}}$ are better than KNN in separating correct and wrong predictions when there are no label-shifted samples, as shown in Fig. 4 (a)&(b). As a result, $RL_{\text{geo-M}}$ and $RL_{\text{conf-M}}$ have the best overall performance when In-D, covariate-shifted and label-shifted samples coexist, as shown in Fig. 4 (d). Also, see Table 3 for numerical confirmation of the above observations, where in all cases $RL_{\text{geo-M}}$ and $RL_{\text{conf-M}}$ are the best or comparable to the best-performing among all score functions. We present the SC results of other `ImageNet` models in Appendix G; our margin-based score functions still stand as the best-performing among all.

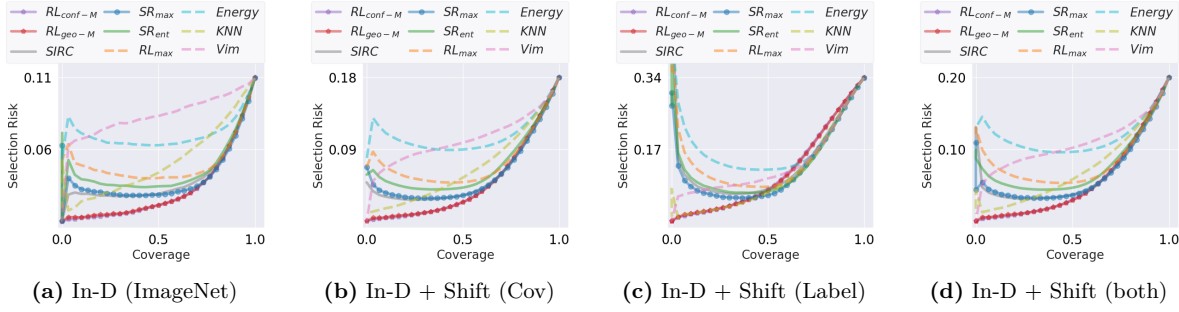

Figure 4: RC curves of different confidence-score functions on the model **EVA** for ImageNet. **(a)-(d)** are RC curves evaluated using samples from **(a)** In-D samples only, **(b)** In-D and covariate-shifted samples only, **(c)** In-D and label-shifted samples only, and **(d)** all samples, respectively. We group the curves by whether they are originally proposed for SC setups (solid lines) or for OOD detection (dashed lines).

Table 3: Summary of AURC-$\alpha$ for Fig. 4. The AURC numbers are *on the $10^{-2}$ scale—the lower, the better*. The score functions proposed for SC are highlighted in gray, and the rest are originally for OOD detection. The best AURC numbers for each coverage level are highlighted in bold, and the $2^{nd}$ and $3^{rd}$ best scores are underlined.

| ImageNet - EVA | In-D | | | In-D + Shift (Cov) | | | In-D + Shift (Label) | | | In-D + Shift (both) | | |
|---|---|---|---|---|---|---|---|---|---|---|---|---|
| $\alpha$ | 0.1 | 0.5 | 1 | 0.1 | 0.5 | 1 | 0.1 | 0.5 | 1 | 0.1 | 0.5 | 1 |
| $RL_{\text{conf-M}}$ | **0.16** | **0.53** | **2.39** | **0.24** | **0.96** | **4.77** | **1.04** | 3.34 | 11.7 | **0.34** | **1.20** | **5.43** |
| $RL_{\text{geo-M}}$ | 0.27 | 0.59 | 2.43 | 0.37 | 1.02 | 4.78 | 1.20 | 3.35 | 11.6 | 0.48 | 1.26 | **5.43** |
| SIRC | 2.23 | 2.07 | 3.36 | 3.71 | 3.06 | 5.83 | 15.8 | 8.88 | 13.7 | 4.61 | 3.53 | 6.52 |
| $SR_{\text{max}}$ | 3.20 | 2.36 | 3.38 | 4.52 | 3.66 | 5.93 | 13.1 | 7.52 | 12.6 | 5.21 | 3.75 | 6.56 |
| $SR_{\text{ent}}$ | 4.28 | 3.13 | 4.04 | 6.24 | 4.66 | 7.00 | 16.0 | 9.19 | 13.4 | 7.04 | 5.10 | 7.61 |
| $SR_{\text{doctor}}$ | 3.22 | 2.38 | 3.40 | 4.55 | 3.40 | 6.00 | 13.2 | 7.55 | 12.6 | 5.24 | 3.78 | 6.61 |
| $RL_{\text{max}}$ | 5.53 | 4.05 | 4.57 | 8.48 | 6.04 | 7.64 | 21.1 | 11.9 | 14.9 | 9.53 | 6.59 | 8.33 |
| Energy | 8.13 | 6.60 | 6.90 | 12.8 | 10.3 | 11.1 | 27.3 | 16.6 | 18.1 | 14.1 | 11.0 | 11.8 |
| KNN | 0.99 | 2.27 | 4.58 | 1.22 | 2.89 | 6.78 | 1.18 | **3.23** | **10.8** | 1.24 | 2.98 | 7.16 |
| ViM | 5.48 | 7.11 | 8.31 | 5.31 | 8.05 | 10.4 | 5.83 | 7.89 | 13.4 | 5.35 | 8.12 | 10.7 |

**Results on iWildCam & Amazon**  We report in Fig. 5 and Table 4 the SC performance of different score functions on `iWildCam` and `Amazon`. Similar to the `ImageNet` experiment above, scores designed for OOD detection ($RL_{\text{max}}$, Energy, KNN and ViM) do not have satisfactory performance in SC. By contrast, existing SR-based scores ($SIRC$, $SR_{\text{max}}$, $SR_{\text{ent}}$ and $SR_{\text{doctor}}$) all demonstrate better SC potential than OOD score functions, and our margin-based score functions ($RL_{\text{conf-M}}$ and $RL_{\text{geo-M}}$) perform on par with the SR-based scores.

## 4.2  Comparison with a training-based confidence-score function

We also compare with a training-based method, ScNet (Geifman & El-Yaniv, 2019). ScNet consists of a selection network and a classifier that are structurally *decoupled* and trained together, allowing us to perform

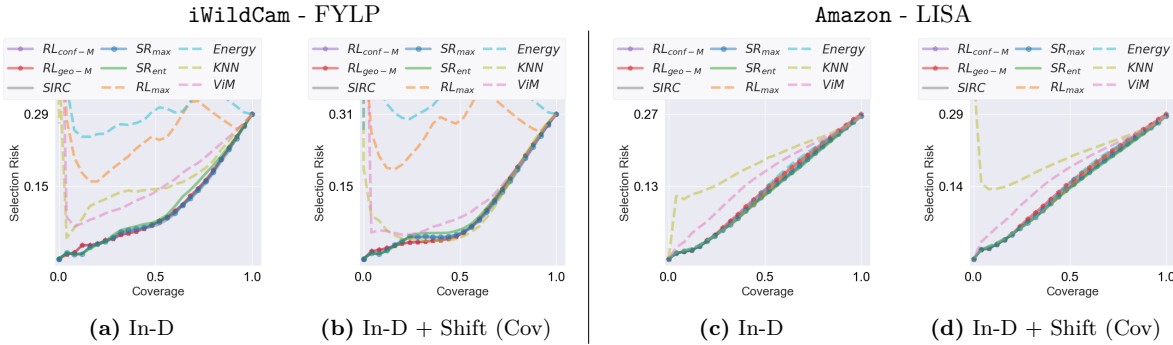

Figure 5: RC curves of different confidence-score functions on the model **FLYP** for `iWildCam` and the model **LISA** for `Amazon`. **(a)&(c)** are RC curves evaluated using In-D samples only and **(b)&(d)** are RC curves evaluated using both In-D and covariate-shifted samples.

Table 4: Summary of AURC-$\alpha$ for Fig. 5. The AURC numbers are *on the $10^{-2}$ scale—the lower, the better.* The score functions proposed for SC are highlighted in gray, and the rest are originally for OOD detection. The best AURC numbers for each coverage level are highlighted in bold, and the $2^{nd}$ and $3^{rd}$ best scores are underlined.

| | iWildCam- FYLP | | | | | | Amazon - LISA | | | | | |
| | In-D | | | In-D + Shift (Cov) | | | In-D | | | In-D + Shift (Cov) | | |
| $\alpha$ | 0.1 | 0.5 | 1 | 0.1 | 0.5 | 1 | 0.1 | 0.5 | 1 | 0.1 | 0.5 | 1 |
|---|---|---|---|---|---|---|---|---|---|---|---|---|
| $RL_{\text{conf-M}}$ | 1.63 | 3.88 | 10.2 | 1.84 | **3.21** | 10.0 | **1.11** | 5.31 | 12.5 | **1.83** | 6.91 | 14.2 |
| $RL_{\text{geo-M}}$ | 1.63 | 3.88 | 10.1 | 1.84 | **3.21** | 10.0 | 1.13 | 5.51 | 12.8 | 1.86 | 7.15 | 14.6 |
| SIRC | **1.45** | **3.72** | **9.84** | 1.38 | 3.5 | **9.94** | 1.14 | 5.09 | 12.2 | 1.88 | 6.66 | 13.9 |
| $SR_{\text{max}}$ | **1.45** | 3.87 | 10.0 | 1.38 | 3.61 | 10.1 | 1.14 | 5.13 | 12.3 | 1.88 | 6.70 | 14.0 |
| $SR_{\text{ent}}$ | 1.46 | 4.03 | 10.6 | **1.34** | 3.94 | 10.6 | 1.15 | **5.06** | **12.1** | 1.89 | **6.61** | **13.8** |
| $SR_{\text{doctor}}$ | **1.45** | 3.87 | 10.1 | 1.38 | 3.62 | 10.1 | 1.14 | 5.13 | 12.2 | 1.88 | 6.70 | 13.9 |
| $RL_{\text{max}}$ | 29.1 | 21.4 | 24.7 | 25.5 | 24.8 | 27.9 | 1.26 | 5.21 | 12.5 | 1.98 | 6.88 | 14.4 |
| Energy | 35.2 | 28.3 | 29.9 | 36.1 | 33.2 | 34.4 | 1.26 | 5.37 | 12.8 | 1.98 | 6.88 | 14.4 |
| KNN | 6.40 | 11.1 | 15.3 | 8.16 | 5.10 | 10.7 | 12.1 | 14.3 | 18.2 | 16.1 | 16.5 | 20.1 |
| ViM | 13.4 | 10.7 | 15.7 | 6.98 | 6.47 | 12.2 | 2.33 | 8.72 | 15.0 | 3.55 | 10.4 | 16.7 |

a faithful comparison of selection scores with a fixed classifier[10]. As shown above, score functions designed for OOD detection perform poorly for generalized SC, so here we focus on comparing our margin-based and SR-based score functions with ScNet. We first train ScNet using the training set of `CIFAR-10` and `ImageNet`, respectively; see Appendix F for training details. After training, we fix both the classification and the selection heads and compute the scores and selection risks using the test setup shown in Table 2: (i) the ScNet selection score is taken directly from the selection head, and (ii) the margin-based and SR-based scores are computed using the classification head.

**Results** We show in Fig. 6 the RC curves achieved using ScNet, SR-based, and margin-based scores. For the `CIFAR` experiment shown in Fig. 6 (a)&(b), ScNet and $RL_{\text{conf-M}}$ perform comparably and are better than $SR_{\text{max}}$ and SIRC, whereas for the `ImageNet` experiment in Fig. 6 (c)&(d), $RL_{\text{conf-M}}$, $RL_{\text{geo-M}}$, $SR_{\text{max}}$ and SIRC perform comparably and are better than ScNet.[11] Surprisingly, ScNet does not always lead to the best

---

[10]We do not consider training-based score functions such as Liu et al. (2019); Huang et al. (2022) due to the ambiguity in calculating their SR responses. During their training, a virtual class "abstention" is added and the softmax normalization is applied on all logits—including that of the virtual class, so it is unfair either simply dropping the abstention logit during test for score calculation or keeping the abstention logit but modifying the score calculation procedure. Retraining a classifier with the same settings but without the abstention logit is also unfair due to the requirement of a fixed classifier. Furthermore, Feng et al. (2023) reports that the above selection methods (Liu et al., 2019; Huang et al., 2022) are not as effective as they claim.

[11]Existing training-based SC works so far have only reported SC (In-D) performance on `CIFAR-10` dataset and have not experimented with `ImageNet` using the full training set. Our results on `CIFAR-10` dataset faithfully reproduce the result originally reported in Geifman & El-Yaniv (2019).

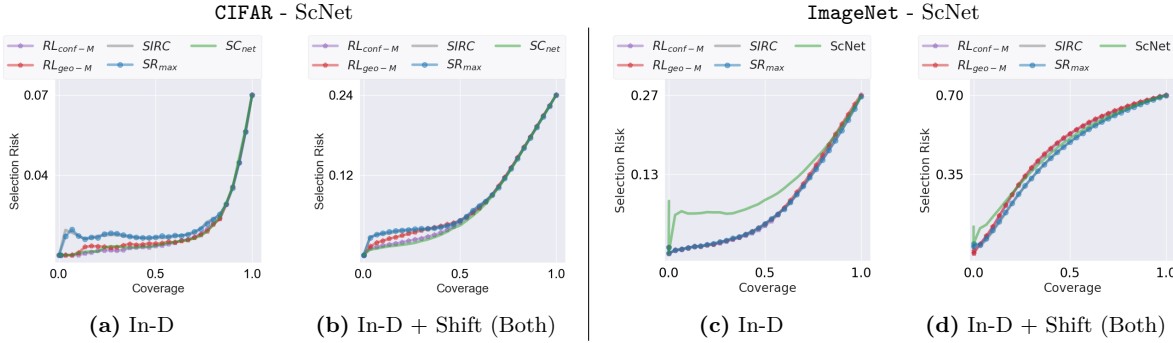

Figure 6: RC curves of different confidence-score functions on the model **ScNet** for `CIFAR` and `ImageNet`. **(a)&(c)** are RC curves evaluated using In-D samples only and **(b)&(d)** are RC curves evaluated using both In-D and covariate-shifted samples.

performance, even if it has access to training data. However, our margin-based scores consistently exhibit good SC performance.

### 4.3 Summary of experimental results

From all above experiments, we can conclude that (i) existing nontraining-based score functions for OOD detection do not perform well for generalized SC, not helping achieve reliable classification performance after rejecting low-confidence samples, and (ii) our proposed margin-based score functions $RL_{\text{geo-M}}$ and $RL_{\text{conf-M}}$ consistently perform comparably to or better than existing SR-based scores on all DL models we have tested, especially in the low-risk regime, which is of particular interest for high-stakes problems. These confirm the superiority of $RL_{\text{geo-M}}$ and $RL_{\text{conf-M}}$ as effective confidence-score functions for SC even under moderate distribution shifts for risk-sensitive applications.

In most of our experiments, $RL_{\text{geo-M}}$ and $RL_{\text{conf-M}}$ perform similarly; only in rare cases, e.g. Fig. 5 (a) and Fig. 6 (b), $RL_{\text{conf-M}}$ slightly outperforms $RL_{\text{geo-M}}$. However, we do not think it is sufficient to conclude that $RL_{\text{conf-M}}$ is better than $RL_{\text{geo-M}}$, or vise versa. Recall how $RL_{\text{conf-M}}$ and $RL_{\text{geo-M}}$ are defined in Eqs. (11) and (12) and their associated decision rules, the current practice of training DL classifiers is in favor of $RL_{\text{conf-M}}$[12]. Thus, understanding the difference in behavior of $RL_{\text{geo-M}}$ and $RL_{\text{conf-M}}$ is likely to also involve investigation of the training process, which we will leave for future work.

## 5 Conclusion and discussion

In this paper, we have proposed *generalized selective classification*, a new selective classification (SC) framework that allows distribution shifts. This is motivated by the pressing need to achieve reliable classification for real-world, risk-sensitive applications where data can come from the wild in deployment. Generalized SC *covers* and *unifies* existing selective classification and out-of-distribution (OOD) detection, and we have proposed two margin-based score functions for generalized SC, $RL_{\text{geo-M}}$ and $RL_{\text{conf-M}}$, which are not based on training: they are compatible for any given pretrained classifiers. Through our extensive analysis and experiments, we have shown the superiority of $RL_{\text{geo-M}}$ and $RL_{\text{conf-M}}$ over numerous recently proposed nontraining-based score functions for SC and OOD detection. As the first work that touches on generalized SC, our paper can inspire several lines of future research, including at least: (i) to further improve the SC performance, one can try to align the training objective with our SC confidence-score functions here, i.e.,

---

[12]The cross-entropy loss is the most commonly used and minimizing it can be viewed as approximating maximizing the confidence margin. To see this, without loss of generality, assume that the magnitudes of the raw logits are ordered $z_1 > z_2 > \cdots > z_K$ and that the true label of the current sample is class 1. Then the cross-entropy loss for the current sample is $-\log\left(e^{z_1}/\sum_i e^{z_i}\right) = \log\sum_i e^{z_i - z_1} = \log\left(1 + \sum_{i \geq 2} e^{z_i - z_1}\right)$, so $\min -\log\left(e^{z_1}/\sum_i e^{z_i}\right) \equiv \min \log\left(1 + \sum_{i \geq 2} e^{z_i - z_1}\right) \equiv \min \sum_{i \geq 2} e^{z_i - z_1}$, where the last minimization problem can be approximated by $\min e^{z_2 - z_1} \equiv \min(z_2 - z_1) \equiv \max(z_2 - z_1)$, i.e., maximizing the confidence margin, when $e^{z_2 - z_1} \gg \sum_{i \geq 3} e^{z_i - z_1}$.

promoting large margins; (ii) in this paper, we only consider the case where all classes are treated equally, while practical generalized SC might entail different rejection weights and costs for different classes, e.g., medical diagnosis of diseases with different levels of health implications; (iii) last but not least, finding better confidence-score functions. We hope that our small step here stimulates further research on generalized SC, bridging the widespread gaps between exploratory AI development and reliable AI deployment for practical high-stakes applications.

### Acknowledgments

Liang H. and Sun J. are partially supported by NIH fund R01NS131314. Peng L. and Sun J. are partially supported by NIH fund R01CA287413. The authors acknowledge the Minnesota Supercomputing Institute (MSI) at the University of Minnesota for providing resources that contributed to the research results reported in this article. The content is solely the responsibility of the authors and does not necessarily represent the official views of the National Institutes of Health. This research is also part of AI-CLIMATE: "AI Institute for Climate-Land Interactions, Mitigation, Adaptation, Tradeoffs and Economy," and is supported by USDA National Institute of Food and Agriculture (NIFA) and the National Science Foundation (NSF) National AI Research Institutes Competitive Award no. 2023-67021-39829.

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

## A  Linear SVM and margins

We first consider binary classification. Assume training set $\{(\boldsymbol{x}_i, y_i)\}_{i \in [N]}$ ($[N] \doteq \{1, \dots, N\}$), where $y_i \in \{+1, -1\}$ and for notational simplicity, we assume that an extra 1 has been appended to the original feature vectors so that we only need to consider the homogeneous form of the predictor: $f(\boldsymbol{x}) = \boldsymbol{w}^\mathsf{T}\boldsymbol{x}$. The basic idea of SVM is to maximize the worst signed geometric margin, *which makes sense no matter whether the data are separable or not*:

$$\max_{\boldsymbol{w}} \min_{i \in [N]} \frac{y_i \boldsymbol{w}^\mathsf{T} \boldsymbol{x}_i}{\|\boldsymbol{w}\|}. \tag{14}$$

Note that the problem is non-convex due to the fractional form $\frac{y_i \boldsymbol{w}^\mathsf{T} \boldsymbol{x}_i}{\|\boldsymbol{w}\|}$. Moreover, $\frac{y_i \boldsymbol{w}^\mathsf{T} \boldsymbol{x}_i}{\|\boldsymbol{w}\|}$ is invariant to the rescaling of $\boldsymbol{w}$, which is bad for numerical computation (as this implies that there exist global solutions arbitrarily close to $\boldsymbol{0}$ and $\infty$).

If the training set is separable, i.e., there exists a $\boldsymbol{w}$ such that $y_i \boldsymbol{w}^\mathsf{T} \boldsymbol{x}_i \geq 0, \forall\, i \in [N]$, there also exists a $\boldsymbol{w}$ so that $\min_i y_i \boldsymbol{w}^\mathsf{T} \boldsymbol{x}_i = 1 \,\forall\, i \in [N]$ by a simple rescaling argument. Then Eq. (14) becomes

$$\max_{\boldsymbol{w}} \min_{i \in [N]} \frac{y_i \boldsymbol{w}^\mathsf{T} \boldsymbol{x}_i}{\|\boldsymbol{w}\|} \quad \text{s.\,t. } \min_i y_i \boldsymbol{w}^\mathsf{T} \boldsymbol{x}_i = 1 \,\forall\, i \in [N] \tag{15}$$

$$\iff \max_{\boldsymbol{w}} \frac{1}{\|\boldsymbol{w}\|} \quad \text{s.\,t. } \min_i y_i \boldsymbol{w}^\mathsf{T} \boldsymbol{x}_i = 1 \,\forall\, i \in [N] \tag{16}$$

$$\iff \min_{\boldsymbol{w}} \|\boldsymbol{w}\| \quad \text{s.\,t. } y_i \boldsymbol{w}^\mathsf{T} \boldsymbol{x}_i \geq 1 \,\forall\, i \in [N], \tag{17}$$

where Eq. (17) is our textbook hard-margin SVM (except for the squared norm often used in the objective). A problem with Eq. (17) is that the constraint set is infeasible for inseparable training data. To fix this issue, we can allow slight violations in the constraint and penalize these violations in the objective of Eq. (17), arriving at

$$\min_{\boldsymbol{w}} \|\boldsymbol{w}\|^2 + C \sum_{i \in [N]} \xi_i \quad \text{s.\,t. } y_i \boldsymbol{w}^\mathsf{T} \boldsymbol{x}_i \geq 1 - \xi_i, \xi_i \geq 0 \,\forall\, i \in [N], \tag{18}$$

which is our textbook soft-margin SVM.

Now for multiclass classification, let us assume the data space: $\mathcal{X} \times \mathcal{Y} = \mathbb{R}^d \times \{1, \dots, K\}$ with $K \geq 3$. The classifier takes the form $f(\boldsymbol{x}) = \boldsymbol{W}^\mathsf{T} \boldsymbol{x}$, where $\boldsymbol{W} \in \mathbb{R}^{d \times K}$. We note that from binary SVM, people create the notion of *confidence margin*:

$$\mathrm{ConfMargin}(\boldsymbol{x}_i, \boldsymbol{w}) \doteq y_i \boldsymbol{w}^\mathsf{T} \boldsymbol{x}_i, \tag{19}$$

which for the binary case is simply the signed geometric margin rescaled by $\|\boldsymbol{w}\|$. The standard multiclass decision rule is[13]

$$\arg\max_{j \in [K]} \boldsymbol{w}_j^\mathsf{T} \boldsymbol{x}, \tag{20}$$

where $\boldsymbol{w}_j$ is the $j$-th column of $\boldsymbol{W}$. To correctly classify all points, we need

$$\forall\, i \in [N],\ y_i = \arg\max_{j \in [K]} \boldsymbol{w}_j^\mathsf{T} \boldsymbol{x} \iff \forall\, i \in [N],\ \boldsymbol{w}_{y_i}^\mathsf{T} \boldsymbol{x}_i > \max_{y \in \mathcal{Y} \setminus \{y_i\}} \boldsymbol{w}_y^\mathsf{T} \boldsymbol{x}_i. \tag{21}$$

This motivates the multiclass hard-margin SVM, separability assumed:

$$\min_{\boldsymbol{W}} \sum_{j \in [K]} \|\boldsymbol{w}_j\|^2 \quad \text{s.t. } \boldsymbol{w}_{y_i}^\mathsf{T} \boldsymbol{x}_i - \max_{y \in \mathcal{Y} \setminus \{y_i\}} \boldsymbol{w}_y^\mathsf{T} \boldsymbol{x}_i \geq 1,\ \forall\, i \in [N], \tag{22}$$

where terms $\boldsymbol{w}_{y_i}^\mathsf{T} \boldsymbol{x}_i - \max_{y \in \mathcal{Y} \setminus \{y_i\}} \boldsymbol{w}_y^\mathsf{T} \boldsymbol{x}_i$ can be viewed as *multiclass confidence margins*, natural generalizations of confidence margins for the binary case. The corresponding soft-margin version is

$$\min_{\boldsymbol{W}} \sum_{j \in [K]} \|\boldsymbol{w}_j\|^2 + C \sum_{i \in [N]} \xi_i \quad \text{s.t. } \boldsymbol{w}_{y_i}^\mathsf{T} \boldsymbol{x}_i - \max_{y \in \mathcal{Y} \setminus \{y_i\}} \boldsymbol{w}_y^\mathsf{T} \boldsymbol{x}_i \geq 1 - \xi_i, \xi_i \geq 0\ \forall\, i \in [N]. \tag{23}$$

Both hard- and soft-margin versions are convex and thus more convenient for numerical optimization.

On the other hand, if we strictly follow the geometric margin interpretation, it seems more natural to formulate multiclass SVM as follows. Consider the decision rule:

$$\arg\max_{j \in [K]} \frac{\boldsymbol{w}_j^\mathsf{T} \boldsymbol{x}}{\|\boldsymbol{w}_j\|}, \tag{24}$$

which would classify all points correctly provided that there exists a $\boldsymbol{W} \in \mathbb{R}^{d \times K}$ satisfying

$$\forall\, i \in [N],\ \frac{\boldsymbol{w}_{y_i}^\mathsf{T} \boldsymbol{x}_i}{\|\boldsymbol{w}_{y_i}\|} > \max_{y \in \mathcal{Y} \setminus \{y_i\}} \frac{\boldsymbol{w}_y^\mathsf{T} \boldsymbol{x}_i}{\|\boldsymbol{w}_y\|}. \tag{25}$$

This motivates an optimization problem on the worst *geometric margins*:

$$\max_{\boldsymbol{W}} \min_{i \in [N]} \left( \frac{\boldsymbol{w}_{y_i}^\mathsf{T} \boldsymbol{x}_i}{\|\boldsymbol{w}_{y_i}\|} - \max_{y \in \mathcal{Y} \setminus \{y_i\}} \frac{\boldsymbol{w}_y^\mathsf{T} \boldsymbol{x}_i}{\|\boldsymbol{w}_y\|} \right). \tag{26}$$

However, this problem is non-convex and thus not popularly adopted.

# B Asymptotic behaviors of $SR_{\mathsf{max}}$, $SR_{\mathsf{doctor}}$, and $SR_{\mathsf{ent}}$

Recall from mathematical analysis that two functions $f(x)$ and $g(x)$ are *asymptotically equivalent* as $x \to \infty$, written as $f(x) \sim g(x)$ as $x \to \infty$, if and only if $f(x) = g(x)(1 + o(1))$ as $x \to \infty$, where $o(\cdot)$ is the standard small-o notation. Note that $f(x) \sim g(x) \iff g(x) \sim f(x)$.

**Lemma B.1.** *Consider the raw logits $\boldsymbol{z}$, and without loss of generality assume that they are ordered in descending order without any ties, i.e., $z^{(1)} > z^{(2)} > \cdots$. We have that as $\lambda \to \infty$,*

$$SR_{\mathsf{max}}(\lambda \boldsymbol{z}) \sim \exp\left(-e^{\lambda(z^{(2)} - z^{(1)})}\right), \quad SR_{\mathsf{doctor}}(\lambda \boldsymbol{z}) \sim 1 - \exp\left(2e^{\lambda(z^{(2)} - z^{(1)})}\right), \quad SR_{\mathsf{ent}}(\lambda \boldsymbol{z}) \sim -e^{\lambda(z^{(2)} - z^{(1)})}.$$

*Moreover, all of the asymptotic functions are monotonically increasing with respect to $z^{(1)} - z^{(2)}$.*

---

[13]The decision rule for the binary case is $\arg\max_{y \in \{+1, -1\}} y \boldsymbol{w}^\mathsf{T} \boldsymbol{x}$. Therefore, we do not need to worry about the $\boldsymbol{w}$'s scaling.

*Proof.* First, for $SR_{\max}$, we have

$$\log SR_{\max}(\lambda\boldsymbol{z}) = \log \frac{e^{\lambda z^{(1)}}}{\sum_i e^{\lambda z^{(i)}}} = -\log \sum_i e^{\lambda(z^{(i)}-z^{(1)})} \sim -\log\left(1 + e^{\lambda(z^{(2)}-z^{(1)})}\right) \tag{27}$$

as $\lambda \to \infty$, because $\sum_{i\geq 3} e^{\lambda(z^{(i)}-z^{(1)})}/(1 + e^{\lambda(z^{(2)}-z^{(1)})}) \to 0$ as $\lambda \to \infty$. Moreover, as $\lambda \to \infty$,

$$e^{\lambda(z^{(2)}-z^{(1)})} \to 0 \implies -\log\left(1 + e^{\lambda(z^{(2)}-z^{(1)})}\right) \sim -e^{\lambda(z^{(2)}-z^{(1)})}, \tag{28}$$

as $\log(1+x) \sim x$ when $x \to 0$. So we conclude that

$$SR_{\max}(\lambda\boldsymbol{z}) \sim \exp\left(-e^{\lambda(z^{(2)}-z^{(1)})}\right) \text{ as } \lambda \to \infty. \tag{29}$$

Now consider $SR_{\text{doctor}}$. Applying a similar argument as above, we have

$$\log \|\sigma(\lambda\boldsymbol{z})\|_2^2 = \log \sum_i \frac{e^{2\lambda z^{(i)}}}{(\sum_j e^{\lambda z^{(j)}})^2} = \log \sum_i \frac{e^{2\lambda(z^{(i)}-z^{(1)})}}{(\sum_j e^{\lambda(z^{(j)}-z^{(1)})})^2}$$

$$= -2\log \sum_j e^{\lambda(z^{(j)}-z^{(1)})} + \log \sum_i e^{2\lambda(z^{(i)}-z^{(1)})} \tag{30}$$

$$\sim -2\log\left(1 + e^{\lambda(z^{(2)}-z^{(1)})}\right) + \log\left(1 + e^{2\lambda(z^{(2)}-z^{(1)})}\right) \tag{31}$$

$$\sim -2e^{\lambda(z^{(2)}-z^{(1)})} + e^{2\lambda(z^{(2)}-z^{(1)})} \tag{32}$$

$$\sim -2e^{\lambda(z^{(2)}-z^{(1)})} \tag{33}$$

as $\lambda \to \infty$, where Eq. (33) holds as $e^{2\lambda(z^{(2)}-z^{(1)})}$ is lower order than $-2e^{\lambda(z^{(2)}-z^{(1)})}$ when $z^{(2)} - z^{(1)} < 0$ so that $e^{\lambda(z^{(2)}-z^{(1)})} < 1$. Therefore, as $\lambda \to \infty$,

$$SR_{\text{doctor}}(\lambda\boldsymbol{z}) = 1 - \|\sigma(\lambda\boldsymbol{z})\|_2^{-2} \sim 1 - \exp\left(2e^{\lambda(z^{(2)}-z^{(1)})}\right). \tag{34}$$

Finally, for $SR_{\text{ent}}$, we have that when $\lambda \to \infty$,

$$SR_{\text{ent}}(\lambda\boldsymbol{z}) = \sum_i \frac{e^{\lambda z^{(i)}}}{\sum_j e^{\lambda z^{(j)}}} \log \frac{e^{\lambda z^{(i)}}}{\sum_j e^{\lambda z^{(j)}}} = \sum_i \frac{e^{\lambda(z^{(i)}-z^{(1)})}}{\sum_j e^{\lambda(z^{(j)}-z^{(1)})}} \log \frac{e^{\lambda(z^{(i)}-z^{(1)})}}{\sum_j e^{\lambda(z^{(j)}-z^{(1)})}} \tag{35}$$

$$= \frac{1}{\sum_j e^{\lambda(z^{(j)}-z^{(1)})}} \sum_i e^{\lambda(z^{(i)}-z^{(1)})}\left(\lambda(z^{(i)} - z^{(1)}) - \log \sum_j e^{\lambda(z^{(j)}-z^{(1)})}\right) \tag{36}$$

$$\sim \frac{1}{\sum_j e^{\lambda(z^{(j)}-z^{(1)})}} \sum_i \left[e^{\lambda(z^{(i)}-z^{(1)})}\lambda(z^{(i)} - z^{(1)})\right] \tag{37}$$

$$\left(\text{as } \log \sum_j e^{\lambda(z^{(j)}-z^{(1)})}/(\lambda(z^{(i)} - z^{(1)})) \in o(1) \text{ when } \lambda \to \infty\right)$$

$$\sim \frac{1}{\sum_j e^{\lambda(z^{(j)}-z^{(1)})}} \left[e^{\lambda(z^{(2)}-z^{(1)})}\lambda(z^{(2)} - z^{(1)})\right], \tag{38}$$

where Eq. (38) holds because $\sum_{i\geq 3} e^{\lambda(z^{(i)}-z^{(1)})}\lambda(z^{(i)} - z^{(1)})/(e^{\lambda(z^{(2)}-z^{(1)})}\lambda(z^{(2)} - z^{(1)})) = \sum_{i\geq 3} e^{\lambda(z^{(i)}-z^{(2)})}(z^{(i)} - z^{(1)})/(z^{(2)} - z^{(1)}) \in o(1)$ as $\lambda \to \infty$. Continuing the above argument, we further have that as $\lambda \to \infty$,

$$\log(-SR_{\text{ent}}(\lambda\boldsymbol{z})) \sim -\log \sum_j e^{\lambda(z^{(j)}-z^{(1)})} + \lambda(z^{(2)} - z^{(1)}) + \log\left(-\lambda(z^{(2)} - z^{(1)})\right). \tag{39}$$

Let's write $x \triangleq -(z^{(2)} - z^{(1)})$. The last two terms in Eq. (39) can be re-written as $-x + \log(x)$. Since $\lim_{x \to \infty} \frac{\log(x)}{x} = 0$, we thus have $-x + \log(x) = -x(1 + o(1))$ as $x \to \infty$, and hense $-x + \log(x) \sim -x$ by the definition of the asymptotic equivalence. Therefore, we have:

$$\log(-SR_{\text{ent}}(\lambda \boldsymbol{z})) \sim -\log\Big(1 + e^{\lambda(z^{(2)} - z^{(1)})}\Big) + \lambda(z^{(2)} - z^{(1)}) \tag{40}$$

$$\sim -e^{\lambda(z^{(2)} - z^{(1)})} + \lambda(z^{(2)} - z^{(1)}) \sim \lambda(z^{(2)} - z^{(1)}). \tag{41}$$

So we conclude that

$$SR_{\text{ent}}(\lambda \boldsymbol{z}) \sim -\exp\Big(\lambda(z^{(2)} - z^{(1)})\Big) \quad \text{as } \lambda \to \infty, \tag{42}$$

completing the proof. $\square$

## C  Evaluation metrics for OOD detection vs. evaluation metrics for generalized SC

The commonly used evaluation metrics for OOD detection do not reflect the classification performance (Franc et al., 2023b). Here we provide a quantitative supporting example, in comparison with the RC curve for generalized SC.

Table 5: Evaluation of $s_1$ and $s_2$ using popular OOD metrics. The better numbers are highlighted in bold.

| OOD metric | $s_1$ | $s_2$ |
|---|---|---|
| AUROC ($\uparrow$) | 0.765 | **0.944** |
| AUPR ($\uparrow$) | 0.987 | **0.997** |
| FPR@TPR=0.95 ($\downarrow$) | 0.816 | **0.279** |

OOD (mostly label-shift) detection as formulated in Eq. (9) can be viewed as a binary classification problem: selected and rejected samples form the two classes. So pioneer work on OOD detection, such as Hendrycks & Gimpel (2016), proposes to evaluate OOD detection in a manner similar to that of binary classification, e.g., using the Area Under the Receiver Operating Characteristic (AUROC) curve (Davis & Goadrich, 2006) and Area Under the Precision-Recall curve (AUPR) (Saito & Rehmsmeier, 2015) to measure the separability of In-D and OOD samples.[14] However, two important aspects are missing in OOD detection, and hence also its performance evaluation, if we are to focus on the performance on the accepted samples:

1. Pretrained classifiers do not always make wrong predictions on label-shifted samples, and hence these OOD samples should not be blindly rejected;
2. In-D samples that might have been correctly classified can be rejected due to poor separation of In-D and OOD samples, leading to worse classification performance on the selected part.

To demonstrate our points quantitatively, we take the pretrained model **EVA**[15] from `timm` (Wightman, 2019) that achieves $> 88\%$ top 1 accuracy on the ImageNet validation set. We then mix `ImageNet` validation set (In-D samples) with `ImageNet-O` (OOD samples, label shifted) (Hendrycks & Dietterich, 2018), and evaluate two score functions $s_1$ and $s_2$[16] using both generalized SC formulation (via RC curves) and OOD detection (via AUROC and AUPR).

According to Table 5, $s_2$ is considered superior to $s_1$ by all metrics for OOD detection. Correspondingly, from Fig. 7(a) and (b), we observe that the scores of the label-shifted samples (green) and those of the In-D samples (blue and orange) are more separated by $s_2$ than by $s_1$. However, we can also quickly notice one issue: In-D samples are not completely separated from OOD samples—a threshold intended to reject label-shifted samples will inevitably reject a portion of In-D samples at the same time, even though a large portion of In-D samples have been correctly classified (blue); In-D samples that can be correctly classified (blue) are less separated from those misclassified ones (orange) by $s_2$ than by $s_1$. This problem cannot be revealed by the OOD metrics in Table 5, but is captured by the RC curves in Fig. 7(c) where the selection risk of $s_2$ (blue) increases as more OOD samples are rejected (TPR from 0.95 to 0.1 as indicated by the

---

[14]A single-point metric, False Positive Rate (FPR) at 0.95 True Positive Rate (TPR), is also popularly used as a companion (Liang et al., 2017; Wang et al., 2022; Liu et al., 2020; Djurisic et al., 2022; Sun et al., 2022; Yang et al., 2022).

[15]See Appendix E for model card information. This model is also used in the experiments of Section 4.

[16]$s_1$ is our proposed $RL_{\text{conf-M}}$ and $s_2$ is ViM.

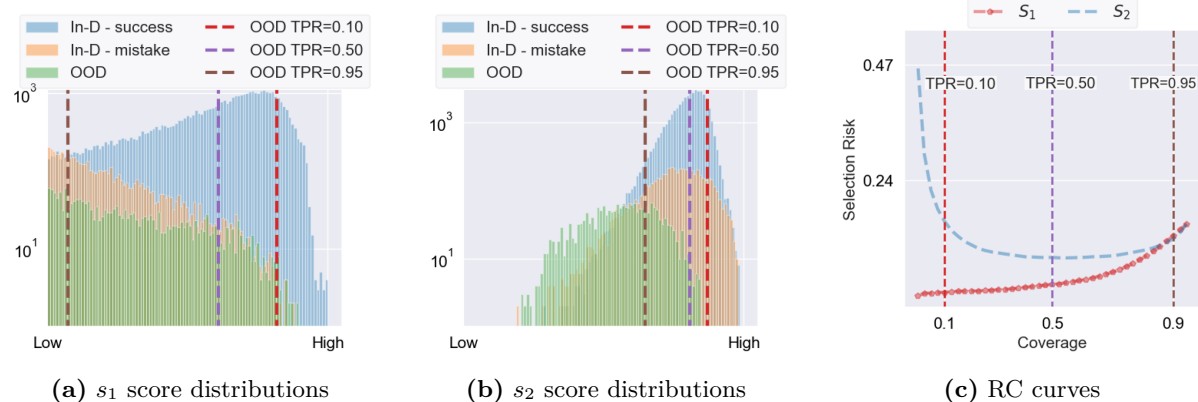

**(a)** $s_1$ score distributions      **(b)** $s_2$ score distributions      **(c)** RC curves

Figure 7: Score distributions of $s_1$ and $s_2$ (a)-(b) and their RC curves (c). In (a) and (b), In-D samples that are *correctly* classified by **EVA** are shown in blue, while In-D samples that are *incorrectly* classified are shown in orange; OOD samples (label-shifted) are shown in green. The vertical dashed lines in (a)-(c) corresponds to different True-Positive-Rate cutoffs in the AUROC metric in OOD detection.

vertical dashed lines). In contrast, the more samples rejected by $s_1$ (smaller coverage), the lower the selection risk, implying that $s_1$ serves SC better.

## D    Rejection patterns of different score functions

We plot in Fig. 8 the heatmap of the score values for each score function. During SC, samples located in the darker areas (with low score values) will be rejected before those located in the brighter areas (with high score values).

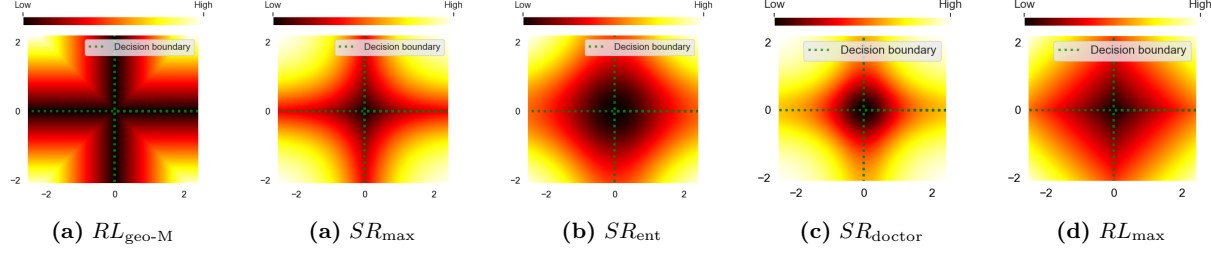

**(a)** $RL_{\text{geo-M}}$     **(a)** $SR_{\text{max}}$     **(b)** $SR_{\text{ent}}$     **(c)** $SR_{\text{doctor}}$     **(d)** $RL_{\text{max}}$

Figure 8: Heatmaps of rejection patterns (distribution of scores). Note that because we rescale the scores for good visualization, the colors are not cross-comparable between different score functions.

## E    Timm model cards

Table 6: Names of model cards in library `timm` to retrieve the models for `ImageNet`

| Dataset | Model name | Model card name | Top-1 Acc. (%) |
|---|---|---|---|
| ImageNet | EVA (ViT) | eva_giant_patch14_224.clip_ft_in1k | 88.76 |
| | ConvNext | convnextv2_base.fcmae_ft_in22k_in1k | 86.25 |
| | VOLO | volo_d4_224.sail_in1k | 85.56 |
| | ResNext | seresnextaa101d_32x8d.sw_in12k_ft_in1k | 85.94 |

Table 6 shows the names of the model cards used to retrieve the pretrained models for `ImageNet` from the `timm` library. Our considerations for choosing these models are as follows: **(i)** the models should cover a

wide range of recent and popular architectures, and **(ii)** they should achieve high top-1 accuracy to represent recent advances of image classification.

## F  Training details for ScNet

We use the unofficial `PyTorch` implementation[17] of the original SelectiveNet (Geifman & El-Yaniv, 2019) due to the out-of-date `Keras` environment of the original repository[18]. The `PyTorch` implementation follows the training method proposed in Geifman & El-Yaniv (2019) and faithfully reproduces the results of `CIFAR-10` experiment reported in the original paper. We add the `ImageNet` experiment on top of the `PyTorch` code, as it is not included in the original code or the paper. Table 7 summarizes the key hyperparameters to produce the results reported in this paper.

Table 7: Key hyperparameters for the ScNet training used in this paper

| Dataset | Model architecture | Dropout prob. | Target coverage | Batch size | Total epochs | Lr (base) | Scheduler |
|---------|--------------------|--------------|-----------------|-----------|--------------|-----------|-----------|
| `CIFAR-10` | VGG | 0.3 | 0.7 | 128 | 300 | 0.1 | StepLR |
| `ImaegNet-1k` | resnet34 | N/A | 0.7 | 768 | 250 | 0.1 | CosineAnnealingLR |

## G  Additional `ImageNet` experiments

We report in Fig. 9 the RC curves of different score functions on models `ConvNext`, `ResNext`, and `VOLO` for `ImageNet`, and summarize their AURC statistics in Table 8.

## H  Ablation experiments for the KNN score

We show in Fig. 10 the SC performance of the KNN score on models `EVA`, `ConvNext`, `ResNext`, and `VOLO`, respectively, on `ImageNet` with all In-D and distribution-shifted samples. We can observe that (i) the SC performance of KNN is sensitive to the choice of hyperparameter $k$, and (ii) our selection $k = 2$ achieves the best SC performance for KNN score on our `ImageNet` task.

---

[17]`https://github.com/gatheluck/pytorch-SelectiveNet`
[18]`https://github.com/anonygit32/SelectiveNet`

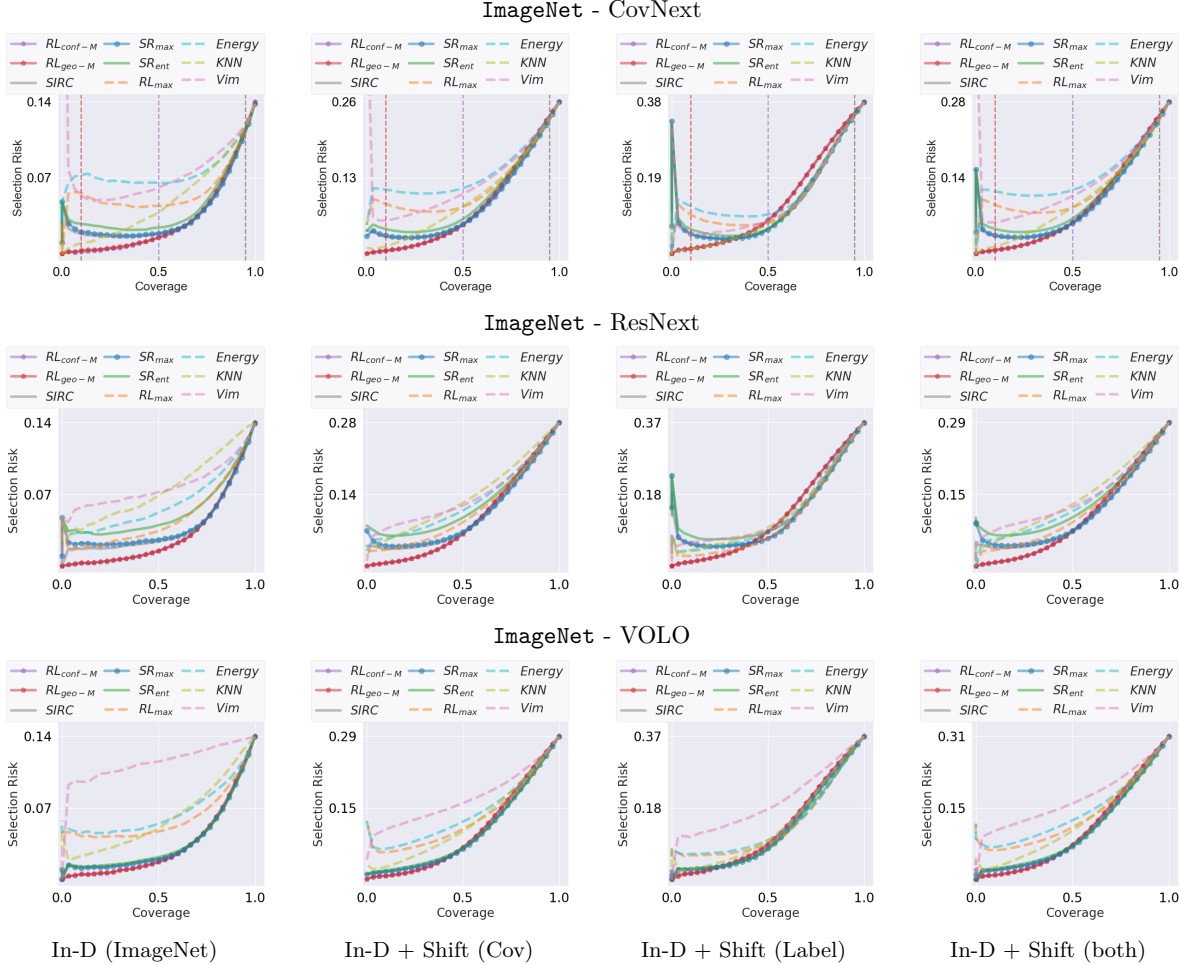

Figure 9: RC curves of different confidence-score functions on models `ConvNext`, `ResNext` and `VOLO` from `timm` for ImageNet. The four columns are RC curves evaluated using samples from In-D only, In-D and covariate-shifted only, In-D and label-shifted only, and all, respectively. We group the curves by whether they are originally proposed for SC (solid lines) or for OOD detection (dashed lines).

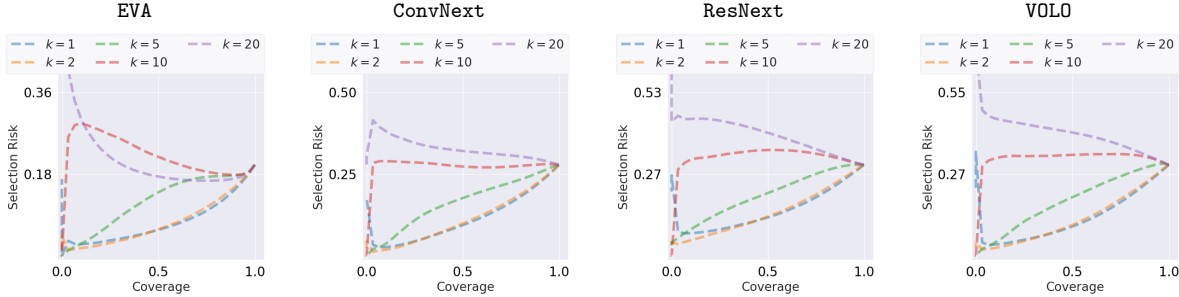

Figure 10: RC curves achieved by the KNN score with different $k$ on `ImageNet`

Table 8: Summary of AURC-$\alpha$ for Fig. 9. The AURC numbers are *on the $10^{-2}$ scale—the lower, the better.* The score functions proposed for SC are highlighted in gray, and the rest are originally for OOD detection. The best AURC numbers for each coverage level are highlighted in bold, and the $2^{nd}$ and $3^{rd}$ best scores are underlined.

| ImageNet - ConvNext | In-D | | | In-D + Shift (Cov) | | | In-D + Shift (Label) | | | In-D + Shift (both) | | |
|---|---|---|---|---|---|---|---|---|---|---|---|---|
| $\alpha$ | 0.1 | 0.5 | 1 | 0.1 | 0.5 | 1 | 0.1 | 0.5 | 1 | 0.1 | 0.5 | 1 |
| $RL_{\text{conf-M}}$ | **0.10** | **0.53** | **3.02** | **0.26** | 1.76 | 8.20 | **0.58** | **2.51** | 11.8 | **0.34** | 1.99 | 8.88 |
| $RL_{\text{geo-M}}$ | 0.15 | 0.59 | 3.10 | 0.31 | **1.75** | **8.14** | 0.75 | 2.54 | 11.8 | 0.38 | **1.97** | **8.81** |
| SIRC | 1.96 | 1.70 | 3.59 | 3.44 | 3.23 | 8.60 | 5.94 | 4.03 | 11.5 | 3.76 | 3.46 | 9.18 |
| $SR_{\text{max}}$ | 2.26 | 1.86 | 3.66 | 3.73 | 3.40 | 8.70 | 5.86 | 4.05 | 11.4 | 4.04 | 3.62 | 9.26 |
| $SR_{\text{ent}}$ | 2.77 | 2.44 | 4.19 | 4.78 | 4.33 | 9.54 | 6.83 | 4.85 | 11.6 | 5.13 | 4.56 | 10.1 |
| $SR_{\text{doctor}}$ | 2.26 | 1.86 | 3.67 | 3.73 | 3.41 | 8.74 | 5.86 | 4.06 | **11.3** | 4.04 | 3.63 | 9.29 |
| $RL_{\text{max}}$ | 5.43 | 4.77 | 5.81 | 9.05 | 7.89 | 11.6 | 10.5 | 7.73 | 13.2 | 9.45 | 8.13 | 12.1 |
| Energy | 6.66 | 6.70 | 7.54 | 10.9 | 10.7 | 13.9 | 11.9 | 9.78 | 14.6 | 11.3 | 10.9 | 14.3 |
| KNN | 1.01 | 2.37 | 5.72 | 1.29 | 4.54 | 10.6 | 1.11 | 3.66 | 12.0 | 1.31 | 4.59 | 11.0 |
| ViM | 15.1 | 9.84 | 9.49 | 16.2 | 11.9 | 14.3 | 14.1 | 9.57 | 14.5 | 16.2 | 11.9 | 14.7 |
| **ImageNet - ResNext** | | | | | | | | | | | | |
| $RL_{\text{conf-M}}$ | **0.12** | **0.59** | **3.17** | **0.29** | 2.15 | 9.38 | **0.59** | 3.22 | 12.8 | **0.38** | 2.50 | 10.2 |
| $RL_{\text{geo-M}}$ | 0.17 | 0.60 | 3.18 | 0.34 | **2.14** | **9.33** | 0.65 | **3.16** | 12.7 | 0.43 | **2.49** | **10.1** |
| SIRC | 1.71 | 1.91 | 3.94 | 3.96 | 4.18 | 9.99 | 7.77 | 5.88 | 13.1 | 4.47 | 4.57 | 10.7 |
| $SR_{\text{max}}$ | 2.28 | 2.26 | 4.11 | 4.88 | 4.69 | 10.3 | 7.44 | 5.88 | 12.9 | 5.36 | 5.06 | 11.0 |
| $SR_{\text{ent}}$ | 3.38 | 3.42 | 5.37 | 6.92 | 6.94 | 12.2 | 9.46 | 7.70 | 13.9 | 7.47 | 7.36 | 12.8 |
| $SR_{\text{doctor}}$ | 2.29 | 2.28 | 4.17 | 4.92 | 4.75 | 10.4 | 7.47 | 5.92 | 12.8 | 5.39 | 5.12 | 11.1 |
| $RL_{\text{max}}$ | 1.57 | 2.34 | 4.79 | 2.98 | 4.82 | 10.9 | 2.37 | 3.83 | **11.9** | 3.06 | 5.00 | 11.4 |
| Energy | 3.08 | 3.90 | 6.17 | 5.13 | 7.20 | 12.7 | 3.68 | 5.34 | 13.2 | 5.19 | 7.37 | 13.2 |
| KNN | 3.23 | 4.84 | 7.61 | 4.12 | 7.65 | 13.6 | 3.40 | 5.85 | 13.5 | 4.14 | 7.77 | 14.0 |
| ViM | 4.68 | 6.13 | 7.79 | 6.18 | 8.81 | 13.6 | 5.09 | 6.82 | 13.6 | 6.23 | 8.92 | 14.1 |
| **ImageNet - VOLO** | | | | | | | | | | | | |
| $RL_{\text{conf-M}}$ | **0.31** | **0.79** | **3.44** | **0.46** | 2.24 | 9.72 | 1.30 | 3.79 | 13.3 | 0.68 | 2.67 | 10.6 |
| $RL_{\text{geo-M}}$ | 0.37 | 0.81 | 3.46 | 0.50 | **2.23** | 9.73 | **0.94** | 3.56 | 13.1 | **0.66** | **2.64** | 10.6 |
| SIRC | 1.27 | 1.44 | 3.74 | 1.35 | 2.82 | 9.56 | 2.68 | 3.97 | 12.9 | 1.90 | 3.37 | 10.5 |
| $SR_{\text{max}}$ | 1.31 | 1.42 | 3.72 | 1.33 | 2.82 | 9.59 | 2.54 | 3.78 | 12.7 | 1.86 | 3.36 | 10.5 |
| $SR_{\text{ent}}$ | 1.47 | 1.59 | 3.83 | 1.58 | 3.13 | 9.72 | 2.71 | 3.87 | **12.4** | 2.13 | 3.69 | 10.6 |
| $SR_{\text{doctor}}$ | 1.31 | 1.42 | 3.71 | 1.33 | 2.82 | **9.55** | 2.54 | 3.78 | 12.7 | 1.86 | 3.36 | **10.4** |
| $RL_{\text{max}}$ | 4.92 | 4.51 | 6.18 | 6.32 | 7.13 | 12.5 | 6.37 | 6.82 | 13.8 | 7.07 | 7.84 | 13.4 |
| Energy | 5.21 | 4.99 | 6.84 | 6.88 | 8.24 | 13.5 | 6.70 | 7.37 | 14.3 | 7.62 | 8.95 | 14.4 |
| KNN | 2.18 | 3.29 | 6.23 | 2.10 | 5.03 | 11.7 | 2.27 | 4.85 | 13.7 | 2.15 | 5.26 | 12.3 |
| ViM | 9.38 | 10.7 | 11.9 | 9.04 | 12.0 | 16.5 | 10.4 | 13.5 | 21.1 | 9.22 | 12.4 | 17.3 |

