# OpenReview forum: "Selective Classification Under Distribution Shifts"
_TMLR — Accepted by TMLR_

### Review · Reviewer_rQhf · 2024-07-09

**Summary Of Contributions:**

This paper introduces a novel framework for selective classification, termed generalized selective classification, which encompasses both label-shifted and covariate-shifted samples. Additionally, the paper presents two margin-based score functions tailored for this task, demonstrating their advantages in experiments compared to previously used score functions.

**Audience:**

Yes

**Broader Impact Concerns:**

No.

**Claims And Evidence:**

Yes

**Requested Changes:**

1. Clearly define the learning setting of generalized SC and provide a comparison to related problems such as OOD detection, OOD generalization, and open-set recognition.
2. Explain why the scaling factor is crucial for the SC problem.
3. Carefully review the paper for mathematical proofs and typos.

**Strengths And Weaknesses:**

##Strengths
- Considering both label-shifted and covariate-shifted samples in the selective classification (SC) problem is practical, and this paper is the first to address this.
- The proposed margin-based score functions are shown to be effective for rejection purposes and are insensitive to the scaling factor.
- The paper is well-written with clear logical organization.
##Weaknesses
- Analyzing the scaling sensitivity of previous score functions is trivial, as this is widely known. A more important question is why scaling sensitivity is crucial for the SC problem. In practice, we often use a standard softmax classifier in deep networks without further scaling.
- When introducing the new setting to SC, it is unclear why rejecting covariate-shifted samples is appropriate, as there are cases where correct classification despite covariate shift is necessary. The differences between generalized SC, open-set recognition, and out-of-distribution (OOD) detection should be clarified.
- On Page 20, Equation (39), $\log(-\lambda(z^{(2)}-z^{(1)}))$ appears to approach $\infty$ as $\lambda \rightarrow \infty$. How is Equation (40) derived from this?

---

> ### Author Response · Authors · 2024-07-29
> **Reply to Reviewer rQhf**
>
> We thank the reviewer for the supportive and insightful comments! Your concerns are addressed below.
>
> > **About the impact of scaling sensitivity of score functions on SC**
>
> We agree that the standard softmax classifier is often used in deep networks without further scaling. Note that for standard classification based on the max-logit rule, rescaling the raw logits and passing them through softmax normalization again will not change the classification performance.
>
> When it comes to SC, the story is entirely different. The scaling sensitivity of the previous score functions themselves, especially those based on softmax responses (SRs), may have been noticed before. Instead, we make the argument in Sec 3.1 that the scaling sensitivity is crucial—or better stated, overlooked—for SC: (1) we show through the numerical experiment around Fig 2 that one can change the SC performance of a given softmax response (SR)-based score function by simply rescaling raw logits and then renormalization through softmax; (2) such sensitivity of SC performance of SR-based score functions to the scale of raw logits implies that one may be able to claim better performance of an SR-based score function by simply rescaling the raw logits—making any SC evaluation without consideration of the logit scaling shaky. This has motivated us to derive the margin-based score functions that are insensitive to the scaling of the raw logits.
>
>
> > **Why reject covariate-shift samples; comparison to related concepts such as out-of-distribution (OOD) detection, OOD generalization, open-set recognition (OSR)**
>
> First, we allow covariate-shifted samples during test/deployment time but do not necessarily reject them. We agree with the reviewer that “there are cases where correct classification despite covariate shift is [present].”  As stated when we introduce generalized SC, the goal of generalized SC is to accept samples correctly classified and reject those misclassified, regardless of which distributions (in-distribution, covariate-shifted or label-shifted) the samples are drawn from; see the paragraph under “Type C errors” in introduction “...we must control the three types of errors, jointly…”. In other words, we hope to maximize the utility of any given classifier, on samples “from the wild”.
>
> In comparison, OOD detection focuses on separating label-shifted and in-distribution samples, without considering the actual classification performance; OOD generalization focuses on correctly classifying in-distribution and covariate-shifted samples; OSR focuses on correctly classifying in-distribution samples, as well as flagging label-shifted samples. By contrast,  generalized SC covers all of in-distribution, label-shifted, and covariate-shifted samples—the widest coverage compared to these related concepts, and targets the most practical and pragmatic metric, classification performance on the selected samples.
>
> We will clarify these points further in our revision.
>
> > **About Eqs. (39) and (40)**
>
> It is correct that $\log(-\lambda(z^{(2)}-z^{(1)}))$ approaches $\infty$ as $\lambda \to \infty$, but the term is lower order than other terms, and hence we can omit it to arrive at Eq. (40). In detail, write $x = -(z^{(2)}-z^{(1)})$ to simplify the notation, and the last two terms of Eq (39) are $-x + \log (x)$. Since $\lim_{x \to \infty} \frac{\log x}{x} = 0$, we have $-x + \log (x) = -x (1+o(1))$ as $x \to \infty$. So, $-x + log (x) \sim -x$ by the definition of the asymptotic equivalence.
>
> We will carefully proofread our mathematical proofs and add sufficient proof details for better readability in our revision.

---

> > ### Comment · Reviewer_rQhf · 2024-07-30
> >
> > Thank you for the responses. I still have questions regarding the scaling factor in the SC problem. If a method could automatically learn the optimal scaling factor, then the sensitivity of score functions becomes less critical for SC. Therefore, in my view, the scaling factor is not a primary concern in SC but rather a weakness of previous methods.

---

> ### Author Response · Authors · 2024-08-01
> **Reply to Reviewer rQhf w.r.t the follow-up concern**
>
> Thank you for being so responsive!
>
> Regarding your concern about the learnability of the optimal scaling factor, to the best of our knowledge, there is no such method yet to learn the optimal scaling considering the joint distribution shifts.
>
> In fact, we think the concern about scaling sensitivity is also related to the concerns raised by **reviewer HfXr**. Thus, we would kindly refer to our **Reply to reviewer HfXr [Part 1] \& [Part 2]** for detailed explanations.

---

> > ### Comment · Reviewer_rQhf · 2024-08-02
> >
> > Actually, I cannot see the other reviewers' comments. I'm not sure why this is the case.

---

> > > ### Author Response · Authors · 2024-08-02
> > > **Reply to Reviewer rQhf [Follow-up]**
> > >
> > > Oh, maybe it is because the third reviewer has not submitted their review yet and not all reviews are available to everyone?
> > >
> > > **[This link](https://docs.google.com/document/d/1T8bp3ADErir3L4YnmjBqED_gCl3YVbCwPvsS9HXlPMs/edit?usp=sharing)** will take you to an anonymous Google Doc, where we have collected the review from **reviewer HfXr** and our reply, for the convenience of our discussion. Everything has been de-identified to make sure that we did not violate the double blind rule.
> > >
> > > Hope this helps.

---

### Review · Reviewer_HfXr · 2024-07-17

**Summary Of Contributions:**

The authors present a method for selective classification (SC) under distribution shifts based on confidence-score functions applied post-training on raw logits in DNN models. Their proposed method for SC uses notions of a confidence margin for selecting samples to reject, and is intended to cover not only in-distribution examples but also label-shifted and covariate-shifted samples. The authors analyze rejection patterns in a toy example which they also use to show that their proposed method does not suffer from the scale invariance issues seen in softmax response-based scores. Empirical performance is also shown on various multiclass classification tasks (mostly image classification as well as one text classification task) and across multiple pre-trained model architectures. Overall, the geometric margin and confidence margin scores they propose are shown to perform comparably or better than existing softmax response-based scores.

**Audience:**

Yes

**Broader Impact Concerns:**

General broader impacts are already described in intro, but it would be good to also discuss implications relating to https://arxiv.org/abs/2010.14134

**Claims And Evidence:**

Yes

**Requested Changes:**

* Figure 3 and its integration with the text could be greatly revised for clarity. The current arrangement seems to be mostly for space efficiency (?). One suggestion -- color-coding different groups of subfigures and matching them with relevant tags in caption and body text could help, if allowed by the journal. I do feel quite strongly about this one, but am willing to be vetoed if others feel this is not a critical change
* Addressing the weaknesses above: I feel the first is quite critical.
* The second weakness is also important to me, but I would be happy with partial coverage. I am especially interested in seeing discussion explicitly addressing the seemingly dramatic differences between ImageNet and non-ImageNet results, where only ImageNet very clearly benefits from the proposed methods

**Strengths And Weaknesses:**

Strengths:

* The authors choose to address an important, practically beneficial, and seemingly underexplored setting for selective classification, where we do not assume that samples are necessarily in-distribution.
* The experiments span a variety of model architectures, as well as two data modalities, which instills confidence in the proposed approaches' generality.
* The proposed methods are applied post-training, which allows for easier adoption


Weaknesses:
* I am not convinced of the authors' claim of novelty. A **[NeurIPS workshop paper](https://openreview.net/forum?id=FiqXqKR26c)** from 2023 seems to have notable overlap, though the submitted work is much wider in scope.
* Limited explanation and analysis of the empirical results. Many unanswered questions about differences in different SC methods' performances across different tasks/datasets and potential impact of aspects of overall task "difficulty"

---

> ### Author Response · Authors · 2024-08-01
> **Reply to reviewer HfXr [Part 1]**
>
> We thank the reviewer for their supportive and insightful comments! Your concerns are addressed below.
>
> >**Concern about the overlap with the [Neurips workshop paper](https://openreview.net/pdf?id=FiqXqKR26c) (NWP)**
>
> We thank the reviewer for pointing out this critical reference that we overlooked in our initial paper submission! Indeed, the very close titles of the two papers (“On selective classification under distribution shift” of the NWP vs. “Selective Classification Under Distribution Shifts” of our paper) could easily suggest their similarities and overlapped scopes. Also, both works argue that softmax-based score functions are broken for selective classification (SC) if the logit scaling is not carefully considered: their SC performance is sensitive to the scaling of the raw logits.
>
> However, after carefully reading and understanding the NWP, we find the following major differences, which we will make sure to carefully clarify the similarities and differences in our revision.
>
> *1. The ideas to fix the scale-sensitivity issue.*
>
>         We propose two margin-based score functions that are invariant to the intrinsic scale of the raw logits.
>
>         NWP proposes post-hoc (perhaps also data-adaptive) optimization of the logit scaling for popular score functions in the literature, based on a held-out calibration set with the same distribution as the training set.
>
> So, the two papers approach the same issue from different angles: (1) NWP needs an additional calibration set to optimize its key hyperparameter(s), compared to ours; requiring such a calibration set may be infeasible for modern pretrained large deep models, for which the training data and distribution are typically not publicly available. So our current setting and solution are much more practical. (2) Moreover, while our emphasis is on the two margin-based score functions, NWP’s emphasis is on their function-agnostic post-doc optimization procedure—thus, their evaluation mostly stresses the performance gain due to their post-doc optimization for any given score function.
>
> *2. The distribution shifts considered.*
>
>         We consider all three distribution categories, i.e., in-distribution, covariate-shifted, and label-shifted samples. All three categories coexist in the test set, and SC performance is evaluated altogether, e.g., Fig. 4 (b-d), striving for the most practical setting.
>
>         NWP considers only two categories, in-distribution and covariate-shifted samples, and SC performance is evaluated for each distribution group separately.
>
> It is worth noting that from Fig 4(a) and (c) in our paper, better SC performance on every coverage level on In-distribution samples only (e.g., RL-geo-M in Fig 4(a)) does not imply a uniform better performance on all coverage levels when the test samples come from mixed distributions (e.g., RL-geo-M in Fig 4(c)). So the separate evaluation used in NWP might not reflect SC performance on realistic data, considering the fact that the results reported by NWP lacks the label-shift category.
>
> *3. The SC performance metrics.*
>
>         To measure SC performance, we only consider the risk-coverage (RC) curve and report (partial-) area-under-RC (AURC) curves under different coverage levels. The reason for reporting such “partial” metrics is that practical scenarios typically only need single operating points on the RC curve, depending on the task requirement, and thus reporting (full-coverage) AURCs only can be misleading. For example, in Fig. 4(c), the RC curves of SR-max and RL-geo-M have a clear crossover; in Table 4 (FYLP, In-D + shift), SIRC is better when coverage is 0.1 and 1.0 and worse when coverage is 0.5 than RL-geo-M.
>
>         NWP mostly measures its SC performance using normalized AURC (NAURC), a normalized variant of AURC that does not directly consider fixed coverage levels as partial AURC metrics we use.

---

> ### Author Response · Authors · 2024-08-01
> **Reply to reviewer HfXr [Part 2]**
>
> > **Addressing the seemingly dramatic differences between ImageNet and non-ImageNet [SC] results, where only ImageNet very clearly benefits from the proposed methods**
>
> Thank you for raising this very interesting point!
>
> First of all, we want to clarify that the AURC and partial AURC metrics that we use to measure SC performance are NOT cross-comparable between different (dataset, classifier) combinations because these metrics depend on the absolute accuracy of the classifier on the said dataset. So, comparing the quantitative performance gaps of different score functions over different datasets can be technically misleading.
>
> Second, even on the same (dataset, classifier) combination, such performance gaps can be easily manipulated by rescaling the raw logits of the SR-based score functions, as we have already demonstrated in Fig 2. For example, in Fig 2, SRmax-4.0 is uniformly closer to RL-conf-M than SRmax-0.1, and our asymptotic analysis in Lemma 3.1 states that when the scale of raw logits approaches infinity, the RC curves of several popular SR-based score functions converge to that of RL-conf-M. To further confirm these claims on real datasets, we have added a similar plot on the iWildCam dataset with the FLYP model. Note that our point here is not to propose a rescaling method to improve the SC performance or claim that our method could beat the best rescaling method: we simply take given pretrained models without any rescaling (calibration), no matter whether the scale of the raw logits is explicitly controlled during the training process.
>
> [*An external link to this iWildCam-FLYP example on an anonymous Google Drive*](https://drive.google.com/file/d/1KmtfFDLYi1rkdT0bcuG1SAb_S8_jUDxR/view?usp=sharing)
>
> Third, talking about rescaling, we want to reiterate why we do not consider the setup in the NWP discussed above. Their rescaling factors are optimized against a calibration (validation) set identically distributed as the training set, which can be hardly available for modern large models. By contrast, our margin-based score functions do not need such calibration sets and thus can be more practical for modern large models.
>
> > **About the reviewer's suggestion to discuss implications related to the notion of fairness introduced in [this paper](https://arxiv.org/abs/2010.14134)**
>
> The accuracy disparity between groups after selection may change and become more complicated when there are distribution shifts, as the margin distribution, which largely determines the disparity, can change substantially when there are distribution shifts. Their original analysis is based on the typical IID assumption. It would be worthwhile studying the disparity in the presence of distribution shifts in the future.
>
> >**About rearranging Fig 3**
>
> Thank you for making this excellent suggestion! Yes, originally, we targeted a 12-page short paper and made the space compromise. Later, we did not expand it in the initial submission. In the revision, we will follow the reviewer’s ideas, plus other means, to make Fig 3 more readable and better integrated with the surrounding text.

---

### Review · Reviewer_YPt5 · 2024-08-13

**Summary Of Contributions:**

This work introduces the concept of generalized selective classification where the objective is to build a classifier and a filtering rule to ensure the best performance tradeoff between classifier accuracy over selected samples and sample coverage. The twist in this work is the authors instead consider a domain adaptation scenario where the objective is to trade off these possibly competing objectives in a test set that may come from a different distribution than training. The considered domain shifts are both at the level of label shift (possibly interacting with a larger label set than during train) or covariate shift. Apparently the main contributions of this work is to introduce two score functions for this setting, $RL_{geo-M}$ and $RL_{conf-M}$ inspired by margin based classification ideas showing existing nontraining-based score functions for OOD detection do not perform well for generalized SC in comparison with these metrics. The authors then support these claims via an extensive experimental evaluation.

**Audience:**

Yes

**Broader Impact Concerns:**

No broader impact concern

**Claims And Evidence:**

Yes

**Requested Changes:**

I would strongly suggest the authors to improve the writing in their manuscript. As is, it is very hard to understand what are the contributions of this work. I understand one of the objectives is to introduce generalized selective classification. It is not clear to me what is the downstream objective here, as it stands it seems to be to select a threshold (or other kind of filtering rule) to ensure the best tradeoff between filtering probability and accuracy. Because of this I don't fully understand the role of the RC curves in all this analysis. If these things could be made clearer I would be more supportive of this paper. I think the contributions may be sufficient to warrant a TMLR publication but I would like to see my concerns addressed.

**Strengths And Weaknesses:**

This work's main asset is the relevance of the problem setting considered in this work for the community. Moving beyond the same train and test setting is important to make ML methods work in practice. That being said, this work is very hard to follow. It is very unclear what the actual contribution is. The problem description is somewhat meandering. I am not sure if this work is proposing new algorithms, new metrics, or how are these metrics used to improve algorithmic performance. This may be just a matter of better writing.

---

> ### Author Response · Authors · 2024-08-23
> **Reply to reviewer YPt5**
>
> We thank the reviewer for their insightful comments in general, and for their appreciation of the value of our generalized selective classification that covers both label shifts and covariate shifts in particular! We address the main concerns raised by reviewer YPt5 below; we will try our best to streamline the paper during revision, and we also welcome the reviewer to raise more specific questions about where the writing gets confusing.
>
> In Sec 2.1, we present the basic framework for the classic SC: a classifier accompanied by a selection/filtering rule that often takes the form of thresholding a confidence score. The generalized SC will take the same form, as stated in Sec 2.3. However, our focus here is not to derive a full-fledged generalized SC algorithm, but to derive effective confidence-score functions—same as what most existing SC or generalized SC papers do, as explained in detail  below.
>
> > **The best risk-coverage tradeoff?**
>
> As we have explained in Sec. 2.5, the goal of this work is not to propose any specific algorithm to select the threshold to achieve the best (selection) risk-coverage tradeoff. Instead, one of our goals is to show that $RL_{geo-M}$ and $RL_{conf-M}$  are the confidence-score functions which have better **potential** to be applied in generalized SC framework. Our introduction of algorithms/optimization formulations to find the best tradeoff, e.g., Alg. 1, Eqs (5) and (6), is only to provide a thorough picture of the related technical background.
>
> > **The role of the RC curve**
>
> Since our goal is to reveal the **potential** of confidence-score functions, the RC curve, which is popularly used in related literature, reveals all possible risk-coverage tradeoffs achievable by all possible thresholds. As we have explained in Sec. 2.4, plotting the RC curves enables us to compare all risk-coverage tradeoffs at different risk/coverage levels achieved by different confidence scores, for the same test set and classifier. Because of this, **a confidence score with better SC potential will present an RC curve leaning towards the lower-right corner of the risk-coverage coordinate**.
>
> That being said, the (partial) area under the RC curve only serves as a single-number summarizing metric of the RC curves for the convenience to compare different confidence-score functions.

---

### Decision · Action_Editor_KPBU · 2024-09-21

**Recommendation:** Accept with minor revision

**Comment:**

The primary concerns revolve around issues of presentation -- writing, clarifying contributions, explaining results, revising figures.  Otherwise, two of three reviewers (the more engaged reviewers) lean in favor of acceptance so these revisions can be verified.

**Audience:**

Researchers with deployed models that may not be easily retrained or those working on settings that require detection of OOD samples.

**Claims And Evidence:**

The goal of the work is to improve model performance under label/covariate-shift -- and specifically to use the same functions for handling multiple types of distribution shift.  The authors provide scoring functions that are compatible with pretrained models (e.g. classifiers based on model confidence) for OOD detection. Results are presented on several standard classification benchmarks with domain shift and on several pretrained models. Results also include RC curves for the confidence-score functions.